# Bispecific BCMA/CD24 CAR-T cells control multiple myeloma growth

Fumou Sun[1,9], Yan Cheng [1,9], Visanu Wanchai [1], Wancheng Guo[1], David Mery[1], Hongwei Xu[1], Dongzheng Gai[1], Eric Siegel [2], Clyde Bailey[1], Cody Ashby [3], Samer Al Hadidi [1], Carolina Schinke [1], Sharmilan Thanendrarajan[1], Yupo Ma[4], Qing Yi [5], Robert Z. Orlowski[6], Maurizio Zangari[1], Frits van Rhee [1], Siegfried Janz [7], Gail Bishop [8], Guido Tricot[1], John D. Shaughnessy Jr[1] & Fenghuang Zhan [1] ✉

Anti-multiple myeloma B cell maturation antigen (BCMA)-specific chimeric antigen receptor (CAR) T-cell therapies represent a promising treatment strategy with high response rates in myeloma. However, durable cures following anti-BCMA CAR-T cell treatment of myeloma are rare. One potential reason is that a small subset of minimal residual myeloma cells seeds relapse. Residual myeloma cells following BCMA-CAR-T-mediated treatment show less-differentiated features and express stem-like genes, including CD24. CD24-positive myeloma cells represent a large fraction of residual myeloma cells after BCMA-CAR-T therapy. In this work, we develop CD24-CAR-T cells and test their ability to eliminate myeloma cells. We find that CD24-CAR-T cells block the CD24-Siglec-10 pathway, thereby enhancing macrophage phagocytic clearance of myeloma cells. Additionally, CD24-CAR-T cells polarize macrophages to a M1-like phenotype. A dual-targeted BCMA-CD24-CAR-T exhibits improved efficacy compared to monospecific BCMA-CAR-T-cell therapy. This work presents an immunotherapeutic approach that targets myeloma cells and promotes tumor cell clearance by macrophages.

While the introduction of numerous innovative therapies over the past two decades has improved outcomes for patients with multiple myeloma (MM), the vast majority of patients still relapse[1,2]. Immunotherapies are changing the treatment landscape for MM and have improved the overall response and survival of patients with relapsed/refractory MM (RRMM)[3,4]. B cell maturation antigen (BCMA) has been targeted by several immunotherapeutic modalities[5,6]. Treatment with chimeric antigen receptor (CAR) T cells revolutionized the treatment of B-cell malignancies and improved the disease control of RRMM[7,8]. Anti-myeloma BCMA-specific CAR T-cell therapies showed high response rates[5,9]. Importantly, the US Food and Drug Administration (FDA) has approved anti-BCMA-specific CAR T-cell therapies for the treatment of MM[10–12].

However, the responses of MM to BCMA-specific CAR T-cell therapies are not durable. The median progression-free survival of Ide-Cel was 8.8 months for the entire study and in the highest dose cohort,

[1]Myeloma Center, Winthrop P. Rockefeller Institute, Department of Internal Medicine, University of Arkansas for Medical Sciences, Little Rock, AR 72205, USA. [2]Department of Biostatistics, University of Arkansas for Medical Sciences, Little Rock, AR 72205, USA. [3]Department of Biomedical Informatics, University of Arkansas for Medical Sciences, Little Rock, AR 72205, USA. [4]iCell Gene Therapeutics LLC, Research & Development Division, Stony Brook, NY 11790, USA. [5]Center for Translational Research in Hematologic Malignancies, Houston Methodist Cancer Center, Houston Methodist Research Institute, Houston, TX 77030, USA. [6]Department of Lymphoma and Myeloma, The University of Texas MD Anderson Cancer Center, Houston, TX 77030, USA. [7]Division of Hematology and Oncology, Department of Medicine, Medical College of Wisconsin, Milwaukee, WI 53226, USA. [8]Department of Microbiology and Immunology, University of Iowa and VA Medical Center, Iowa City, IA 52242, USA. [9]These authors contributed equally: Fumou Sun, Yan Cheng. ✉e-mail: FZhan@uams.edu

the median progression-free survival was 12.1 months[5,10,13]. One known mechanism of relapse has been traced to the loss of BCMA expression following CAR-T therapy[14]. Dormant, drug-resistant, minimal residual MM cells can also cause relapses[15,16]. There is a strong correlation between the presence of minimal residual MM cells and drug resistance and relapse[17,18], suggesting that minimal residual MM cells-targeted therapies could improve outcomes. Residual myeloma cells following BCMA-CAR T-mediated treatment show less-differentiated features and enrichment of the epithelial–mesenchymal transition (EMT) pathway[19]. EMT is associated with features of stemness[20]. We previously demonstrated that MM cells expressing CD24 exhibit features of drug resistance and self-renewal[17]. In a mouse model of MM, drug-resistant dormant MM cells express a myeloid transcriptome signature that includes CD24[21]. Li et al. employed single-cell RNA sequencing (scRNA-seq) to delineate cell populations within CD45[+] bone marrow cells of patients with RRMM both before and after BCMA CAR-T treatment. Their findings indicated that CD24 was found to be expressed in myeloma cells at relapse following BCMA CAR-T cell therapy[22].

In the tumor microenvironment, CD24 is an important checkpoint molecule controlling the innate immune response[23,24]. CD24 on tumor cells binds to sialic-acid-binding Ig-like lectin 10 (Siglec-10) on macrophages causing immune-cell inhibition, which is mediated by Src homology 2 domain-containing protein tyrosine phosphatase 1/2 (SHP-1/2)[25,26]. These phosphatases are associated with immune receptor tyrosine inhibitory motifs (ITIMs), which are present in the cytoplasmic tail of Siglec-10. When phosphorylated, the ITIM region blocks Toll-like receptor (TLR)-mediated inflammation and activates a series of intracellular signaling pathways promoting tumor escape from immune surveillance. The phosphorylated ITIM region also causes inhibition of cytoskeletal rearrangement, blocking macrophage phagocytic clearance[27]. For these reasons, the CD24-Siglec-10 axis may represent a target for immune therapy.

In this work, we generate CAR-T cells that recognize the CD24 antigen on minimal residual MM cells and can block the "don't eat me" CD24-Siglec-10 pathway inducing tumor cell clearance by phagocytic macrophages. Inhibition of the CD24-Siglec-10 pathway also leads to the activation of inflammatory signaling and anti-tumor signaling. This immunotherapeutic modality may promote more durable remissions in MM.

## Results

### CD24-positive MM cells increase after BCMA-CAR-T treatment
To study the changes in the myeloma cell transcriptome in patients undergoing BCMA-CAR-T therapy (clinical trial NCT02546167), we systematically analyzed bone marrow (BM) scRNA-seq data of MM patients from Emory University (GSE210079)[19] (Fig. 1a). The Uniform Manifold Approximation and Projection (UMAP) plot identified 11 cell clusters based on genetic markers (Fig. 1b). The marker gene UMAP showed the expression of markers from different cell clusters (Supplementary Fig. 1). Split UMAP plot (Fig. 1c) and bar views of the quantitative proportion clusters (Fig. 1d) presented BM cells derived from the pre-infusion group ($n = 5$) and post-infusion Day 28 (D28) group ($n = 6$). As expected, the percentage of MM cells declined on Day 28 post BCMA-CAR-T infusion in patients, when compared with pre-infusion samples (mean pre-infusion 29.5% vs. D28 0.07% $P = 0.005$). However, minimal residual MM cells cause relapse. Comparison of transcriptional profiles of pre-infusion MM with post-infusion MM revealed that residual MM cells showed decreased expression of mature plasma cell genes (such as syndecan-1/*SDC1*, BCMA/*TNFRSF17*, and *XBP1*), but increased expression of the stem-like marker, *SOX4*, and drug-resistant associated makers, including *CD24* (Fig. 1e). Based on the violin plot, it is evident that the proportion of CD24-positive MM cells in the post-infusion D28 group significantly increases when compared to the pre-infusion group. (3.8% vs 24.1%, $P < 0.001$) (Fig. 1f).

Taking patient 19 as a representative example, the gene expression heatmap of MM cells showed that this patient had an IgGλ MM, the proportion and intensity of CD24-positive cells was remarkably increased in residual IgG and λ MM cells after BCMA-CAR-T therapy (Supplementary Fig. 2). Pathway analysis of differential gene expression (DGE) revealed enrichment of the hematopoietic pathway and epithelial–mesenchymal transition (EMT)-associated pathways (cell adhesion molecules, tight junction, and leukocyte transendothelial migration) in residual MM cells (Fig. 1g). These results confirm that CD24 may be a potential CAR-T target to prevent resistance to BCMA-CAR-T treatment.

We collected bone marrow mononuclear cells (BMMCs) from 35 patients with monoclonal gammopathy of undetermined significance (MGUS) and smoldering multiple myeloma (SMM), 39 patients with newly diagnosed MM (NDMM), and 21 patients after treatment with RRMM. Using flow cytometry on BMMCs, we found that the percentage of CD24[+]/CD138[+]/CD38[+]/CD45[−] plasma cells was significantly increased in RRMM (17.00% ± 9.31%) compared with patients with MGUS/SMM (8.00 ± 4.90%) and NDMM patients (12.14% ± 8.00%; Fig. 1h; $P < 0.001$). We treated 15 MM patients' BMMCs with 5 nM bortezomib (BTZ) for 24 h. The frequency of CD138[+]CD24[+] cells increased in 12 of 15 BMMC samples post-BTZ (Fig. 1i). Meanwhile, the other ten MM patients' BMMCs were treated with 10 μM dexamethasone (DEX) or 10 μM lenalidomide (LEN) for 24 h. The frequency of CD138[+]CD24[+] cells increased in eight of ten BMMC samples post-DEX ($P = 0.03$) (Supplementary Fig. 3a). However, the frequency of CD138[+]CD24[+] cells did not change significantly in BMMC samples post-LEN ($P = 0.85$) (Supplementary Fig. 3b). These results indicate that CD24[+] MM cells appear not to be sensitive to bortezomib and dexamethasone. We detected CD24 expression in 20 MM cell lines. We found CD24 is broadly expressed in all MM cell lines. We used the MM cell lines OPM2 (28.3% cells are CD24[+]), MM1.S (11.7% cells are CD24[+]), and H929 (7.77% cells are CD24[+]) for our in vivo and in vitro experiments. (Supplementary Fig. 4)

### Construction and characteristics of CD24-CAR-T cells
A second-generation CD24-CAR vector was constructed by CD24-specific single-chain variable fragments (scFvs; derived from anti-CD24 antibody, SWA11)[28,29], a safety switch (marker-suicide gene, RQR8)[30] in the hinge region, and a 4-1BB co-activation domain with CD3ζ. To decrease the risk of severe immunological side effects, we integrated RQR8 with two CD20 mimotopes as a suicide molecule and one CD34 epitope for detection (Fig. 2a). We modified the CD3[+] T cells from healthy donors' peripheral blood mononuclear cells (PBMCs)[31]. The transduction efficacy of CAR-T cells was detected by flow cytometry. The transduction efficacy of CD24-CAR-T cells was 43% (Supplementary Fig. 5a). CD34-positive CAR-T cells were enriched to 83.6% using CD34 microbeads (Supplementary Fig. 5b). The results of the proliferation assay showed CAR-T cells proliferated well in vitro (Supplementary Fig. 5c). The phenotype of CAR-T cells, including the ratio of CD4 and CD8 phenotypes and the ratio of different memory phenotypes, was detected by flow cytometry. The mean percentage of CD4 CD24-CAR-T cells was 38.9% and CD8 was 53.9%. The combined percentage of CD4 stem cell memory (41.2%) and central memory (21.5%) was more than 60%. CD8 stem cell memory (17.7%) and central memory (43.6%) were also prominent (Fig. 2b). Previous studies found that early memory phenotypes (stem cell memory and central memory) drive T-cell proliferation, persistence, and antitumor effects[32,33].

### CD24-CAR-T cells show efficient MM cells killing in vitro
Analysis of coculture-killing assays of CAR-T cells with OPM2, MM1.S, and H929 indicated that effector CAR-T cell-mediated target tumor cytotoxicity proportionally increased with the ratio of effector: target (E:T) in these 3 MM cell lines. At the ratio 5:1 of CD24-CAR-T versus MM cells, lysed MM cells were 78.9%, 65.0% and 38.7% in the OPM2, MM1.S

 

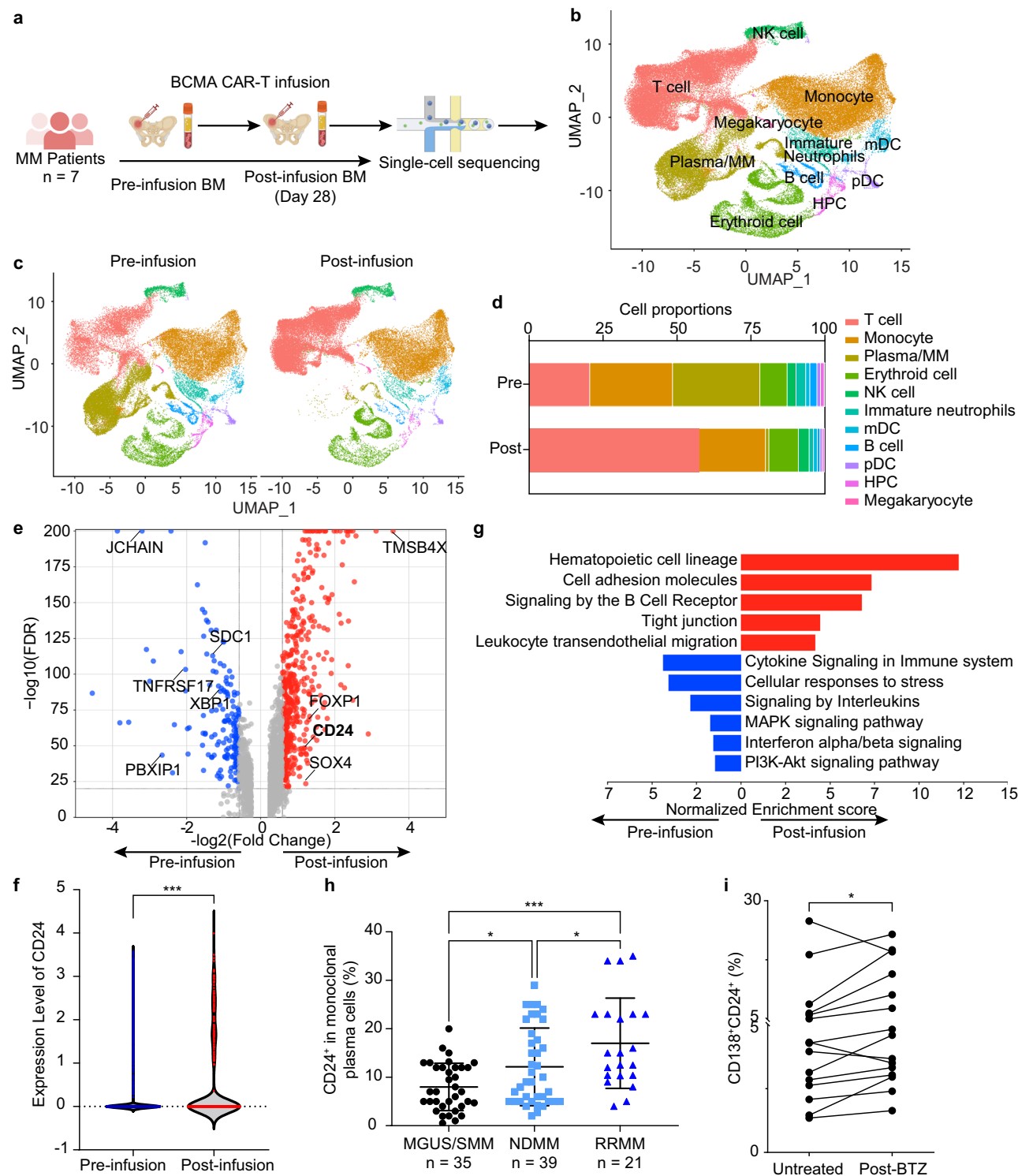

and H929 cell lines, respectively (Fig. 2c). CD24-CAR-T cell activation was confirmed by increased CD69 expression (Fig. 2d). Compared with the MOCK-CAR-T group, interferon (IFN)-γ, interleukin (IL)-2, and tumor necrosis factor (TNF)-α productions in conditioned media were dramatically increased in the CD24-CAR-T group (Fig. 2e, f and Supplementary Fig. 6a). These results suggest that exposure to CD24⁺ MM cells resulted in a strong activation of CD24-CAR-T cells. When the efficacy of CD24-CAR-T was tested in primary human BMMC samples, the percentage of CD138⁺ cells significantly decreased after the CD24-CAR-T treatment (Fig. 2g). These results showed that primary MM cells were effectively lysed by the CD24-CAR-T cells.

To assess the antigen specificity and effector capacity of CD24-CAR-T cells, CAR T-cells were co-cultured with CD24-overexpressing MM cell lines (ARP1$^{CD24OE}$ and OCI-MY5$^{CD24OE}$) and CD24-negative human embryonic kidney cell line (HEK293). When the ratio of CAR-T: target was 5:1, target cells were lysed 99%, 89% and 1.5% for ARP1$^{CD24OE}$, OCI-MY5$^{CD24OE}$ and HEK293, respectively (Supplementary Fig. 6b, c). In the presence of ARP1$^{CD24OE}$ and OCI-MY5$^{CD24OE}$ cell lines, CD24-CAR-T cells released high levels of IFN-γ, IL-2, and TNF-α; but only background cytokine levels were detected when challenged with HEK293 cells (Supplementary Fig. 6d). Furthermore, we employed Clustered regularly interspaced palindromic repeats (CRISPR)-Cas9

**Fig. 1 | Proportion of CD24-positive cells is increased in MM after treatment.**
**a** Schematic approach for myeloma patients' bone marrow single-cell RNA sequencing analysis. **b** UMAP plot of whole bone marrow mononuclear cells (BMMCs). Whole BMMCs were divided into 11 subclusters: T cell, Monocyte, Plasma/MM, B cell, Erythroid cell, Immature Neutrophils, Myeloid dendritic cell (mDC), Plasmacytoid dendritic cell (pDC), Hematopoietic progenitor cell (HPC) and Megakaryocyte. **c** UMAP plot of BMMCs derived from Pre-infusion ($n = 5$) and Post-infusion Day 28 ($n = 6$). **d** Bar-views showed the proportion of various cell types in BMMCs of Pre-infusion ($n = 5$) and Post-infusion Day 28 ($n = 6$). **e** Volcano plot of the top differentially expressed genes (DEGs) in residual MM cells. **f** The violin plot of CD24 expression in each pre-infusion MM and post-infusion D28 MM cells ($n = 5$ independent patient samples with 13,108 Plasma/MM cells in pre-infusion group; $n = 6$ independent patient samples with 460 Plasma/MM cells in

post-infusion group). **g** Pathway analysis of the top DEGs in residual MM cells. **h** The percentage of CD24$^+$ monoclonal plasma cells were compared in samples from patients with monoclonal gammopathy of undetermined significance/smoldering MM (MGUS/SMM) ($n = 35$), patients with newly diagnosed MM (NDMM) ($n = 39$), and patients after treatment with relapsed/refractory MM (RRMM) ($n = 21$). The mean percentage of CD24$^+$/CD138$^+$/CD38$^+$/CD45$^-$ plasma cells was increased after treatment. One-way ANOVA was used. Data are presented as mean values ± SD. **i** Flow cytometry analysis of patient samples and the association between drug response and CD24 expression in MM cells ($n = 15$). The frequency of the sub-population of CD138$^+$CD24$^+$ cells increase in 12 of 15 primary myeloma samples post-bortezomib (BTZ) treatment. Paired $t$ test was used. All tests are two-sided. $^*P < 0.05$, $^{***}P < 0.001$. Raw data are provided in the Source Data file. Exact $P$ values for each comparison shown in (**f**) and (**h**–**i**) can be found in Supplementary Data 1.

technology to knock out CD24 in OPM2 and MM1.S cells (Supplementary Fig. 7a). Subsequently, the cytotoxicity of CD24-CAR-T cells against these knockout MM cells was tested. When the ratio of CD24-CAR-T cell: MM cell was 5:1, 13.7% of target cells were lysed in the OPM2$^{CD24KO}$ cell line, a slight increase compared to the MOCK-CAR-T group (9.58%), that was not statistically significant. Similarly, the same phenomenon occurred when MM1.S$^{CD24KO}$ cell line was used as target cells (Supplementary Fig. 7b). Meanwhile, CD24-CAR-T cells also didn't show the highest activation when OPM2$^{CD24KO}$ and MM1.S$^{CD24KO}$ cell lines were used as target cells (Supplementary Fig. 7c). Compared to the T cells and MOCK-CAR-T cells, CD24-CAR-T cells exhibited a slight increase in the production of IFN-γ, IL-2, and TNF-α, but the differences were not even significant (Supplementary Fig. 7d). These results further substantiate the specificity of CD24-CAR-Tcells targeting.

To further verify the specificity and cross-reactivity of the CD24-CAR-T cells, cells were incubated with fluorescein isothiocyanate (FITC)-labeled human CD24, mouse CD24 and human BCMA, separately. After incubation of FITC-labeled human CD24 and mouse CD24, CD24-CAR-T cells expressed a strong FITC signal. In contrast, there was no FITC signal detected in the BCMA group (Supplementary Fig. 8). These results indicate that CD24-CAR-T cells specifically target human CD24 and showed cross-reactivity with mouse CD24.

## CD24-CAR-T cells promote phagocytic clearance by macrophages

CD24-CAR-T cells were incubated with macrophages, stained with DiD (1,1-dioctadecyl-3,3,3,3-tetramethylindodicarbocyanine; red), and OPM2 stably expressing green fluorescent protein (GFP) to detect the phagocytic clearance of MM cells. Phagocytosis events were visualized as green cells inside the red cells (DiD$^+$GFP$^+$) by flow cytometry. We found that compared with the PBS and MOCK-CAR-T groups, CD24-CAR-T cells showed a strong increase in macrophage phagocytic clearance (Fig. 2h, i).

## CD24-CAR-T cells reduce MM burden in the 5TGM1 mouse model

We performed flow cytometry and detected CD24 positivity in 29.1% of 5TGM1 MM cells using an anti-mouse CD24 antibody, and SWA11 antibody showed similar recognition and binding to CD24$^+$ 5TGM1 cells (Supplementary Fig. 9a). We then constructed mouse CD24-CAR-T cells using the SWA11 scFv and 5TGM1 cells were effectively lysed by mouse CD24-CAR-T cells (Supplementary Fig. 9b). We tested the efficacy of the mouse CD24-CAR-T cells in the immune-competent 5TGM1/C57BL/KaLwRij mouse model (Fig. 3a). We divided the mice into four groups: 1) normal group (healthy mice without 5TGM1 injection), 2) the PBS group (treated with PBS after 5TGM1 injection), 3) MOCK-CAR-T group (treated with MOCK-CAR T after 5TGM1 injection), and 4) CD24-CAR-T group (treated with CD24-CAR T after 5TGM1 injection). Serum protein electrophoresis (SPE) results showed the M protein of the PBS group and the MOCK-CAR-T group increased substantially when compared with the normal group, while the M protein of the CD24-CAR-T group was significantly reduced

compared with the PBS group and the MOCK-CAR-T group (Fig. 3b). 5TGM1 cells (GFP$^+$ cells) decreased more than eightfold lower in the CD24-CAR-T group (4.2%) than in the PBS (35.4%) and MOCK-CAR T (35.0%) groups (Fig. 3c). These results indicate that the CD24-CAR-T can reduce tumor burden in the 5TGM1 model.

Dormant MM cells showed increased CD24 expression[21]. To detect whether CD24-CAR-T cells can eliminate dormant MM cells, 5TGM1-GFP MM cells were stained with DiD and injected into KaLwRij mice[21]. Dormant MM cells were identified by expression of GFP and retention of DiD (GFP$^+$DiD$^{Hi}$). Activated cells that had proliferated and diluted the DiD label were GFP$^+$DiD$^{Neg}$. The results showed that dormant MM cells (GFP$^+$DiD$^{Hi}$) were significantly decreased in the CD24-CAR-T group compared with the MOCK-CAR-T group (Supplementary Fig. 9c) indicating that CD24-CAR-T cells can deplete 5TGM1 dormant cells in vivo. Phagocytosed MM cells were also evaluated from these mice. The ratio of phagocytosis was represented by GFP$^+$ macrophages because 5TGM1 cells were GFP$^+$. CD11b and F4/80 double-positive macrophages from mouse BM were analyzed by flow cytometry; the percentage of GFP$^+$ macrophages was significantly higher in the CD24-CAR-T group than that in the PBS and MOCK-CAR-T groups (Fig. 3d). These findings suggested that the CD24-CAR-T cells enhance phagocytic clearance of tumor cells in vivo.

In the macrophage (CD11b$^+$F4/80$^+$) population, CD86 is used as a marker of M1-like macrophages (M1Φ), while CD206 is used as the marker of M2-like macrophages (M2Φ). Previous studies have shown that M1Φ are involved in the clearance of tumor cells and the M2Φ are an immunosuppressive macrophage[34]. After CD24-CAR-T treatment, we found a significant change in the ratios of M1Φ and M2Φ cells in the myeloma microenvironment. Compared with the normal group, M1Φ cells were reduced by more than 1.7-fold in the PBS group and the MOCK-CAR-T. Conversely, M2Φ cells were increased by greater than 5.5-fold, indicating the ability of tumor cells to drive macrophages toward an M2Φ phenotype. After CD24-CAR-T treatment, this phenomenon was reversed. M1Φ increased greater than twofold compared to the normal group, while M2Φ returned to normal levels (Fig. 3e, f). These data indicate that tumor cells polarize macrophage to a tumor-permissive phenotype and that CD24-CAR-T can reverse this phenomenon.

Importantly, we also found that the median survival was more than 60% longer in the CD24-CAR-T group (42 days) than in control groups (26 days each; $P = 0.0018$; Fig. 3g), with no significant decrease in mice weight after treatment with CD24-CAR-T cells (Supplementary Fig. 9d). These data show that CD24-CAR-T can eliminate tumor cells in vivo and improve the outcome of MM-bearing mice.

## CD24-CAR-T cells modulate myeloma microenvironment

We next analyzed the immune cell changes in the BM microenvironment of the 5TGM1 mouse model using scRNA-seq. The UMAP plot identified 16 cell clusters based on genetic markers (Supplementary Fig. 10a). The dot plot showed the expression of markers from different cell clusters (Supplementary Fig. 10b). Split UMAP plot (Fig. 4a) and

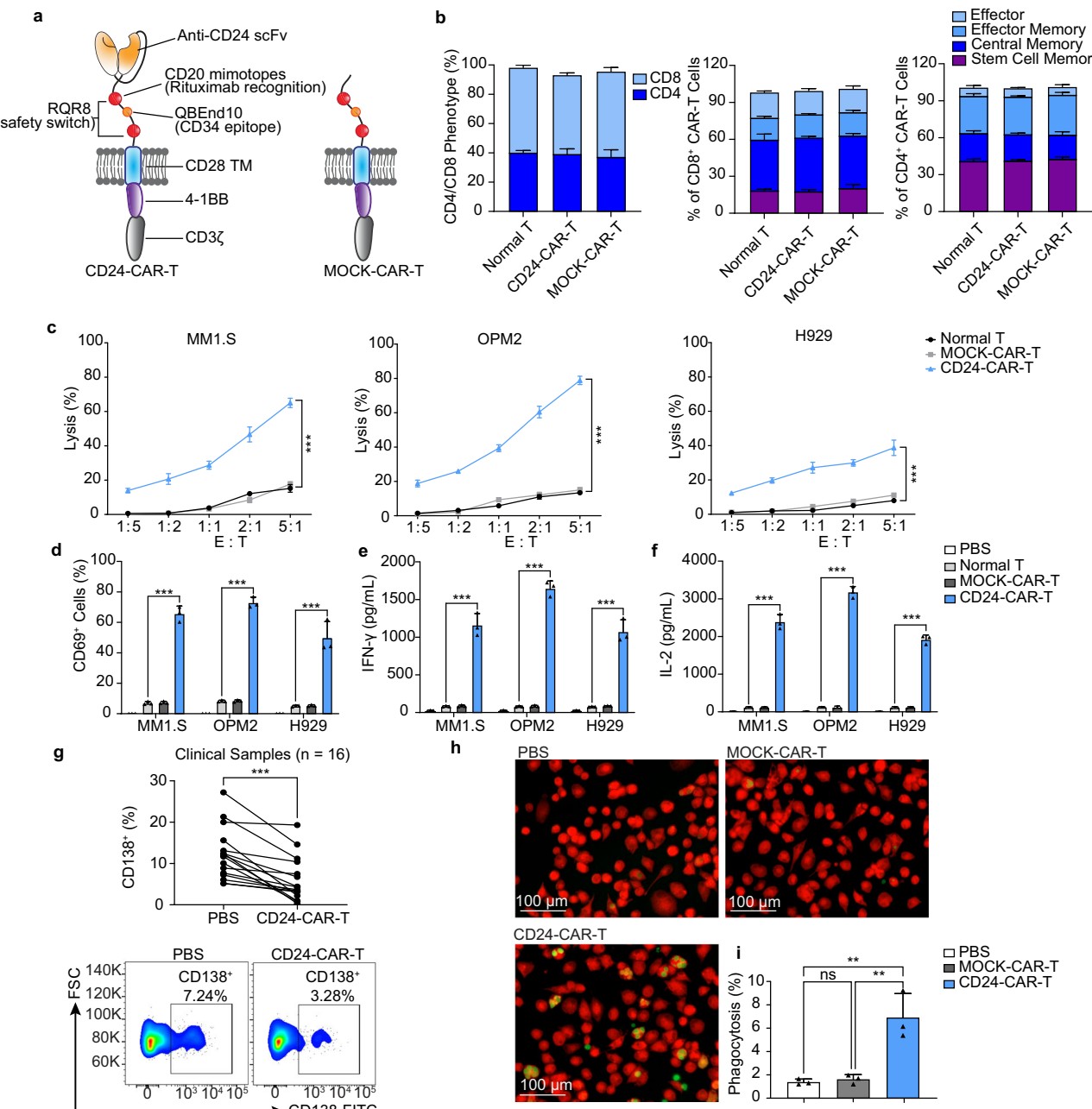

**Fig. 2 | CD24-CAR-T cells target MM cells and promote macrophage phagocytic clearance in vitro. a** CD24-CAR-T and MOCK-CAR-T constructs. CD24-CAR vector was constructed by CD24-specific single-chain variable fragments (scFvs), a safety switch (RQR8), and a 4-1BB co-activation domain with CD3ζ. The MOCK-CAR vector contains a safety switch and a 4-1BB co-activation domain with CD3ζ. **b** The phenotype of CAR-T cells, including the ratio of CD4 and CD8 phenotypes and the ratio of different memory phenotypes (*n* = 3 independent experiments). Stem cell memory (CD45RO⁻/CD62L⁺), central memory (CD45RO⁺/CD62L⁺), effector memory (CD45RO⁺/CD62L⁻), effector cells (CD45RO⁻/CD62L⁻). **c** CD24-CAR-T cells cytolytic activity in vitro. CAR-T or T cells were added to MM1.S, OMP2, and H929 cell lines at the effector/target (E/T) ratio from 1:5 to 5:1. After 24 h of coculture, cytolytic activity was measured (*n* = 3 independent experiments). **d** The expression of T cell activation marker CD69 (*n* = 3 independent experiments). Data are presented as mean values ± SD. **e** Interleukin (IL)-2 concentrations on supernatants (*n* = 3 independent experiments). **f** Interferon (IFN)-γ concentrations on supernatants (*n* = 3

independent experiments). **g** MM patient samples with CD24-CAR-T cells or PBS treatment for 24 h (*n* = 16). The percentage of the subpopulation of CD138⁺ cells decreased in 16 of 16 primary myeloma samples post-CD24-CART treatment. A representative example of flow cytometry analysis. **h** Representative fluorescent images of phagocytic clearance. Phagocytosis was performed by coculture of OPM2 cells that expressed GFP (green), DiD-stained macrophages (red), and CAR-T cells at a ratio of 2:1:1. After a 4-h coculture, suspended cells were washed and detected (*n* = 3 independent experiments). The experiment was repeated twice with the same results. **i** Bar plot showing the percentage of phagocytosis detected by flow cytometry analysis (*n* = 3 independent experiments). Data are presented as mean values ± SD in (**b**–**f**) and (**i**). One-way ANOVA was used in (**c**–**f**), (**i**) and paired t-test was used in (**g**). All tests are two-sided. **P < 0.01, ***P < 0.001, ns = P > 0.05. Raw data are provided in the Source Data file. Exact *P* values for each comparison shown in (**c**, **d**) and (**f**, **g**) can be found in Supplementary Data 1.

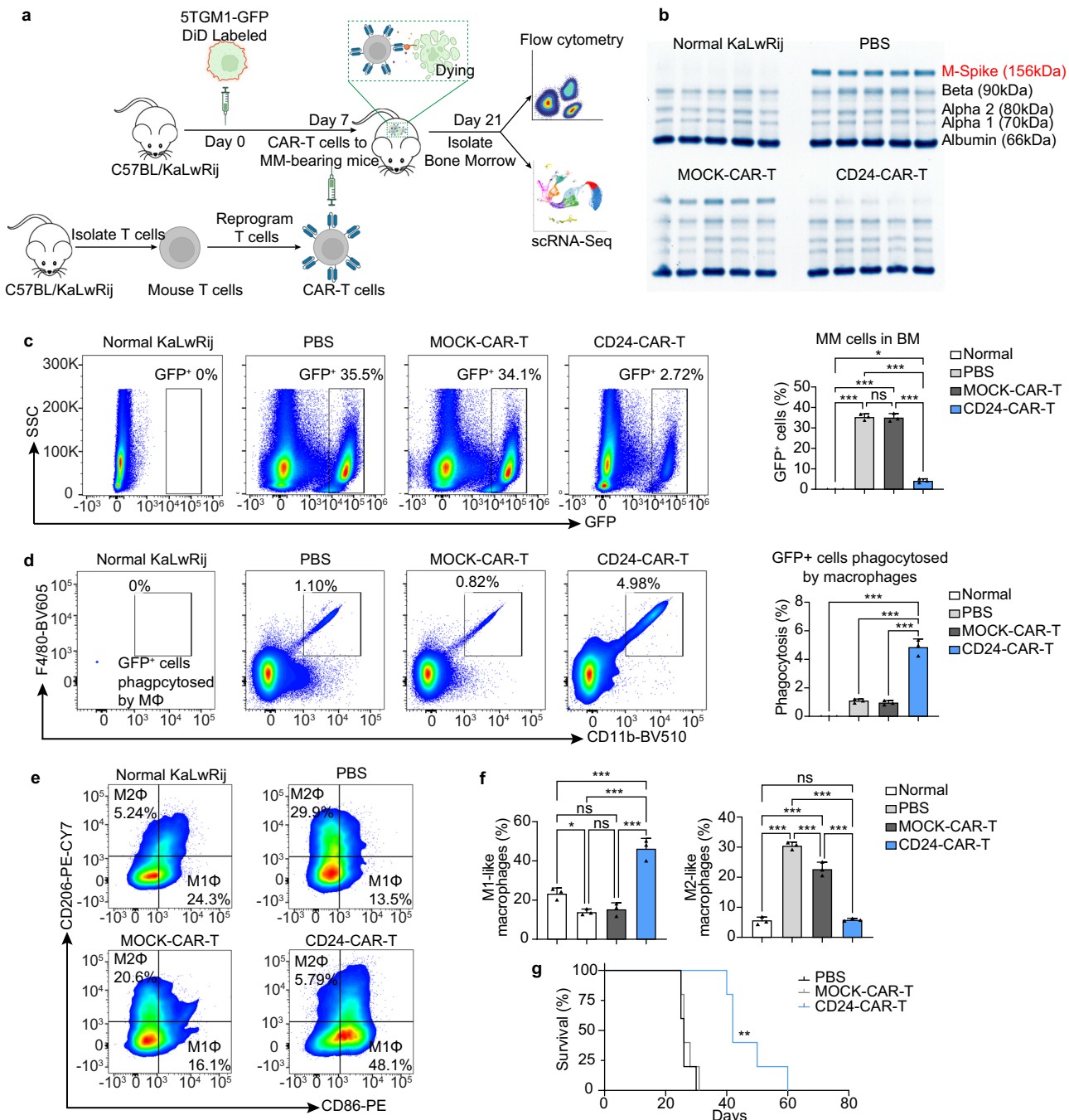

**Fig. 3 | CD24-CAR-T cells target MM cells and promote macrophage phagocytic clearance in vivo. a** Schematic approach for 5TGM1 mouse model. C57BL/KaLwRij mice were intravenously injected with either PBS or 5TGM1-GFP-DiD cells. After 7 days after injection of 5TGM1 cells, mice were treated with either PBS, MOCK-CAR-T cells, or CD24-CAR-T cells. On day 21 after 5TGM1-cell inoculation, mice were killed. Serum electrophoresis (SPE) was performed. Bone marrow mononuclear cells (BMMCs) were isolated. Dormant (GFP⁺DiD^Hi) and activated (GFP⁺DiD^Neg) cells were detected by flow cytometry. BM microenvironmental cells were sorted out for single-cell RNA sequencing (scRNA-seq) (*n* = 5 mice per group). **b** SPE of 5TGM1 models. The M-spike is indicated in red (*n* = 5 per group). The gel has been cut from the outside; no samples/bands were removed. The experiment was repeated twice with the same results. **c** Representative GFP⁺ gating strategy to identify MM cell populations in 5TGM1 BM samples. Bar plot showing the percentage of MM cells

(GFP⁺ cells) in 5TGM1 BM samples (*n* = 3 per group). **d** Representative F4/80⁺CD11b⁺ gating strategy to identify the population of MM cells phagocytosed by macrophages. Bar plot showing the percentage of GFP⁺ cells phagocytosed by macrophages (*n* = 3 per group). **e** Representative CD206⁺ and CD86⁺ gating strategy to identify M2-like-phenotype and M1-like-phenotype cell populations in 5TGM1 BM samples (*n* = 3 per group). **f** Bar plot showing the percentage of M1-like-phenotype and M2-like-phenotype macrophages after treatment (*n* = 3 per group). **g** Kaplan-Meier survival analysis of CAR-T treatment in 5TGM1 models (*n* = 5 per group). Data are presented as mean values ± SD in (**c**, **d**) and (**f**). One-way ANOVA was used for statistical analysis. *$P < 0.05$, **$P < 0.01$, ***$P < 0.001$, ns = $P > 0.05$. Raw data are provided in the Source Data file. Exact $P$ values for each comparison shown in (**c**–**f**), (**g**), and (**i**) can be found in Supplementary Data 1.

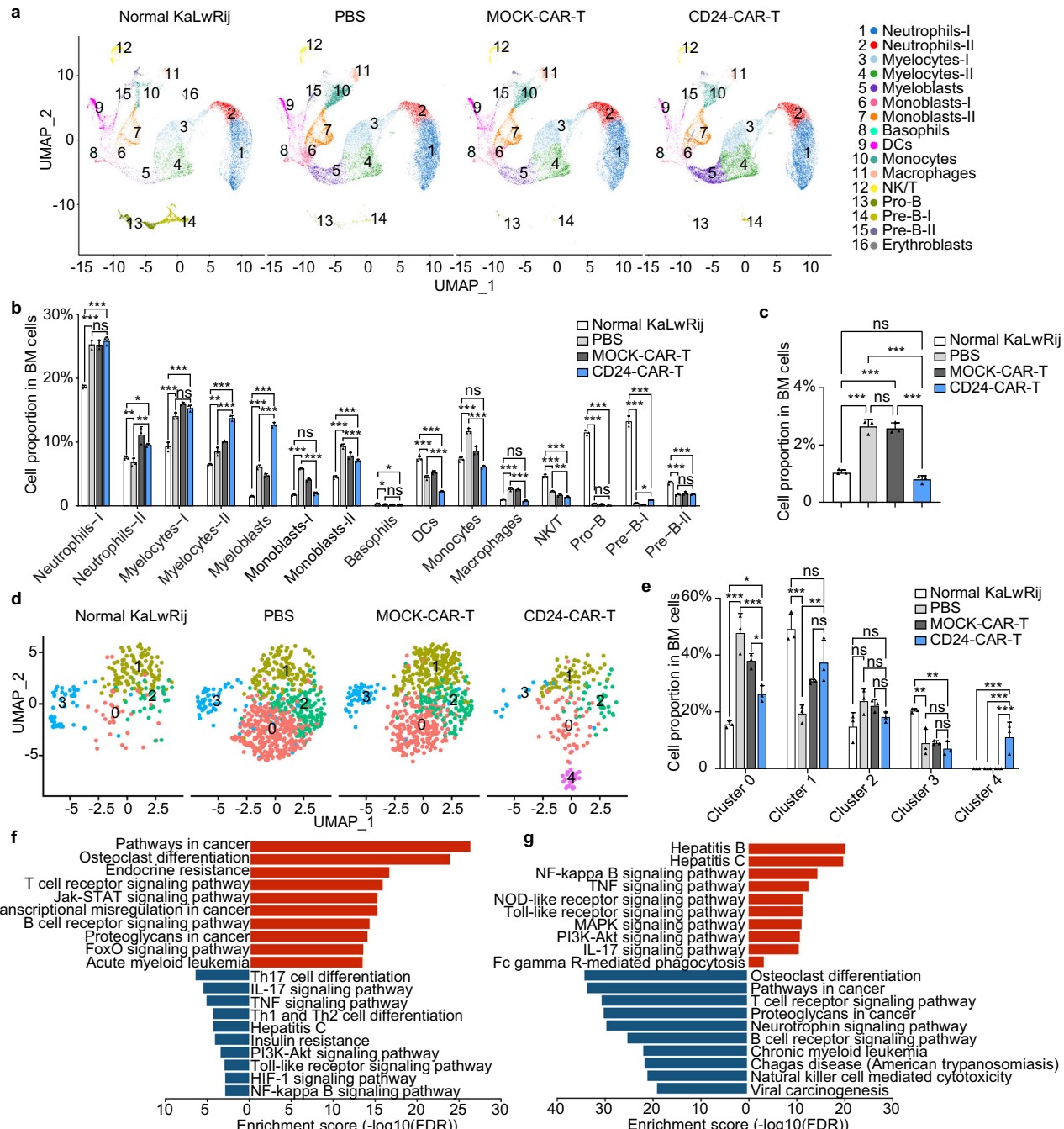

**Fig. 4 | scRNA-seq revealed tumor microenvironment modification after CD24-CAR-T cells treatment. a** UMAP plot of BMMCs derived from healthy mice ($n = 3$ mice per group), MM-bearing mice treated with PBS ($n = 3$ mice per group), MOCK-CAR-T ($n = 3$ mice per group) or CD24 CAR-T ($n = 3$ mice per group). Whole BMMCs were divided into 16 subclusters (Neutrophils-I, Neutrophils-II, Myelocytes-I, Myelocytes-II, Myeloblasts, Monoblasts-I, Monoblasts-II, Basophils, dendritic cells [DCs], Monocytes, Macrophages, Natural killer/ T cells [NK/T], Pro-B, Pre-B-I, Pre-B-II, Erythroblasts). **b** Bar-views showed the proportion of various cell types in BMMCs of healthy and MM-bearing mice treated with PBS, MOCK-CAR-T, or CD24-CAR-T cells ($n = 3$ mice per group). **c** Bar-views showed the proportion of

macrophages in whole BMMCs ($n = 3$ mice per group). **(d)** UMAP plot of macrophages subcluster. Macrophages were divided into five subclusters. **e** Bar-views showed the proportion of cells in subclusters of macrophages of healthy and MM-bearing mice treated with PBS, MOCK-CAR-T, or CD24-CAR-T cells ($n = 3$ mice per group). **f** KEGG pathway analysis of dramatically changed subcluster 0 of macrophages. **g** KEGG pathway analysis of dramatically changed subcluster 4 of macrophages. Data are presented as mean values ± SD in (**b, c**) and (**e**). One-way ANOVA was used for statistical analysis. $^*P < 0.05$, $^{**}P < 0.01$, $^{***}P < 0.001$, ns = $P > 0.05$. Raw data are provided in the Source Data file. Exact $P$ values for each comparison shown in (**b, c**) and (**e**) can be found in Supplementary Data 1.

bar views of the quantitative proportion clusters (Fig. 4b and Supplementary Fig. 11) of BM cells derived from the normal group ($n = 3$), PBS group ($n = 3$), MOCK-CAR-T group ($n = 3$), or CD24-CAR-T group ($n = 3$) revealed several interesting cell cluster changes among the four groups. The series of neutrophil clusters (including its progenitors' myelocytes and myeloblasts clusters) increased in 5TGM1-bearing

mice and continued to rise after CD24-CAR-T treatment. The series of macrophage clusters (including its progenitors' monoblast and monocyte clusters) was increased by more than 2.5-fold in tumor-bearing mice but was decreased to 80% of normal levels after treatment with CD24-CAR-T cells (Fig. 4c). Dendritic cells and natural killer (NK)/T cells declined in tumor-bearing mice and continued to decline

after CD24-CAR-T cells treatment. Importantly, B cells decreased in tumor-bearing mice and increased after CD24-CAR-T cell treatment.

CD24-CAR-T cells clearly affect macrophages (Fig. 4c). We analyzed the proportion of cells in subclusters of macrophages and discovered that two subclusters were dramatically altered in mice treated with CD24-CAR-T cells compared to the controls (Fig. 4d, e). In comparison to the normal group, there is a significant increase in the proportion of cells in Cluster 0 in both the PBS group and the MOCK group. However, in the CD24-CAR-T treatment group, the proportion of cells in Cluster 0 is significantly reduced compared to the PBS group and MOCK group, although it has not yet returned to the levels seen in the normal group. Cluster 4 was a new cluster that appeared after CD24-CAR-T cells treatment. We identified differentially expressed genes between these two clusters and compared those with the other clusters. Pathway analysis was performed for these genes separately. Most of the signaling pathways that are upregulated in Cluster 0 are tumor-associated or osteoclast-associated. Downregulated signaling pathways are associated with inflammation (Fig. 4f). Most of the signaling pathways that are upregulated in Cluster 4 are inflammation- or phagocytosis-associated. Downregulated signaling pathways are tumor- or osteoclast-associated (Fig. 4g).

To further validate the impact of CD24-CAR-T cells on the CD24/Siglec-10 signaling at the protein level, we collected proteins from macrophages in the 5TGM1 mouse models and conducted Western blot analysis to assess the phosphorylated protein levels of SHP-1 and SHP−2. In comparison to the normal group, the levels of phosphorylated SHP-1 and SHP-2 proteins were significantly elevated in both the PBS group and the MOCK group. However, in the CD24-CAR-T treatment group, the levels of phosphorylated SHP-1 and SHP−2 proteins were notably reduced compared to the PBS group and MOCK group, although they had not completely returned to the levels observed in the normal group (Supplementary Fig. 12). This suggests that CD24-CAR-T cells exert a significant inhibitory effect on the phosphorylation of SHP-1 and SHP-2 proteins within the CD24/Siglec-10 signaling pathway. These results matched our previous results: CD24-CAR-T cells block the CD24-Siglec10 ("don't eat me" signal) pathway that allows macrophages to phagocytize tumor cells, and they also activate inflammation-associated signaling pathways. Due to the inflammatory reaction, the series of neutrophils (myelocytes, myeloblasts) was increased after CD24-CAR-T treatment.

Through cell-cell communication analysis of the CD24-CAR-T treatment group using scRNAseq, we found that the highest number of interactions occurred within monocytes and monoblasts, and interactions between monocytes and monoblasts. Regarding interaction weights or strength, the most prominent interactions were observed within neutrophils and between neutrophils and myelocytes (Supplementary Fig. 13a, b). In the CD24-CAR-T treatment group, the most pronounced communication signals were related to the Galectin-9 (Lgals9) signaling pathway. The most critical sources and targets of these signals were neutrophils and monocytes (Supplementary Fig. 13c). It has been reported that Galectin-9 is upregulated in murine M1-like macrophages[35], which aligns with our flow cytometry results indicating a significant increase in M1-like macrophages after CD24-CAR-T treatment. We found in the PBS and MOCK-CAR-T treatment groups, that the Siglec-G (known as Siglec-10 in Humans) signaling pathway showed a significant enhancement (Supplementary Fig. 13d). In contrast, in the CD24-CAR-T treatment group, this signal did not significantly enhance. The interaction between CD24 and Siglec-G is crucial for activating the "don't eat me" signal pathway[25]. CD24-CAR-T cells can effectively inhibit the binding of CD24 to Siglec-G, thereby inhibiting the "don't eat me" signal.

To determine if the increase in inflammation caused a cytokine storm that can result in tissue damage to mice, we tested the spleen, liver, and kidney of the four groups with hematoxylin and eosin (H&E) staining. Compared to the normal group, the CD24-CAR-T group did not experience appreciable organ damage (Supplementary Fig. 14).

## Bispecific CAR-T cells exhibit increased cytolytic activity

We constructed three types of bispecific BCMA-CD24-CAR T cells to target both the bulk tumor cells and drug-resistant myeloma cells and further improve the antitumor efficacy of CAR-T cells: one dual-targeted CAR-T cell (Bi-CAR-T) and two tandem CAR-T cells (Fig. 5a). CD4/CD8 ratio and memory T-cell percentages were analyzed. The results showed that the proportion of CD4 Bi-CAR-T cells was 37.6% and CD8 was 53.8%, and the total percentage of CD4 stem cell memory (42.3%) and central memory (22.7%) was more than 60%. CD8 stem cell memory (17.5%) and central memory (44.1%) also occupied a large proportion of the T-cells (Fig. 5b).

Due to the high expression levels of BCMA in MM1.S (96.6%), OPM2 (91.3%), and H929 (99.5%) cell lines (Supplementary Fig. 15a), CAR-T cells targeting BCMA have demonstrated high MM cell lysis capabilities. The cytolytic activity results showed that compared with other groups, the Bi-CAR-T cells had the highest cytolytic activity in vitro. When the ratio of Bi-CAR-T cell: MM cell was 5:1, 87.3% of target cells were lysed in the MM1.S cell line, and there was a 1.3-fold increase in lysis of MM cells with Bi-CAR-T cells compared to CD24-CAR-T cells. In the OPM2 cell line, 94.2% of target cells were lysed, and there was a 1.3-fold increase in lysis of MM cells with Bi-CAR-T cells compared to CD24-CAR-T cells. In the H929 cell line, 93.0% MM cells were lysed by Bi-CAR-T cells, and there was a 2.1-fold increase compared to the CD24-CAR-T cells (Fig. 5c). Bi-CAR-T cells also showed the highest activation (Fig. 5d). Compared with other groups, Bi-CAR-T cells produced the highest levels of IFN-γ, IL−2, and TNF-α (Fig. 5e, f; Supplementary Fig. 15b). For primary human MM samples, the percentage of the subpopulation of CD138+ cells decreased by more than 42% in 16 of 16 primary myeloma samples after Bi-CAR-T treatment (Fig. 5g).

Compared with the MOCK-CAR-T cells and BCMA-CAR-T cells, the Bi-CAR-T cells strongly promoted phagocytic clearance by macrophages, similar to the CD24-CAR-T cells (Fig. 5h and Supplementary Fig. 16a, c). Flow cytometry also showed the Bi-CAR-T cells significantly increased the phagocytic clearance by macrophages (Fig. 5i and Supplementary Fig. 16b, d). Because the Bi-CAR-T cells were superior to either of the tandem CAR-T cells with respect to in vitro activity, we decided to focus on Bi-CAR-T cells for our in vivo experiments.

## Bispecific BCMA-CD24-CAR-T cells prolong MM mouse survival

To determine CAR-T cells killing human MM cells in vivo, immunodeficient NOD.Cg-*Prkdc<sup>scid</sup> Il2rg<sup>tm1Wjl</sup>*/SzJ (NSG) mice were injected human MM cell lines MM1.S and OPM2 and subsequently treated with CAR-T cells or PBS (Fig. 6a). Bioluminescence imaging of MM1.S-bearing mice revealed Bi-CAR-T cells were the most effective in terms of antitumor activity, yielding near-complete tumor clearance by day 21 (Fig. 6b). Notably, animals treated with CD24-CAR-T cells had better tumor clearance than those treated with MOCK-CAR-T cells, but not as good as those treated with Bi-CAR-T cells (Fig. 6c). Additionally, median survival was more than 75% longer in the Bi-CAR-T group (63 days) than it was in the MOCK-CAR-T groups (36 days; $P = 0.0021$; Fig. 6d). Consistently, Bi-CAR-T cells showed the best antitumor activity and the best survival in the OPM2-bearing mice. (Fig. 6e−g). Furthermore, in the MM1.S-xenograft mouse model, both the PBS and MOCK-CAR-T groups of mice experienced a significant decline in body weight in the later stages of the experiment due to tumor growth. In contrast, the Bi-CAR-T group showed a steady increase in mouse body weight. In the OPM2-xenograft mouse model, the body weight of mice in the MOCK-CAR-T group did not increase, while the Bi-CAR-T group exhibited a steady increase in body weight. These results suggest that Bi-CAR-T cells, while inhibiting MM growth, do not induce significant treatment-related toxicity (Supplementary Fig. 17).

## Discussion

BCMA-targeting CAR-T therapy has become an effective therapeutic approach for MM[5]. FDA-approved BCMA-CAR-T cell therapy has shown impressive overall response rates[10,12]. However, the durability of these

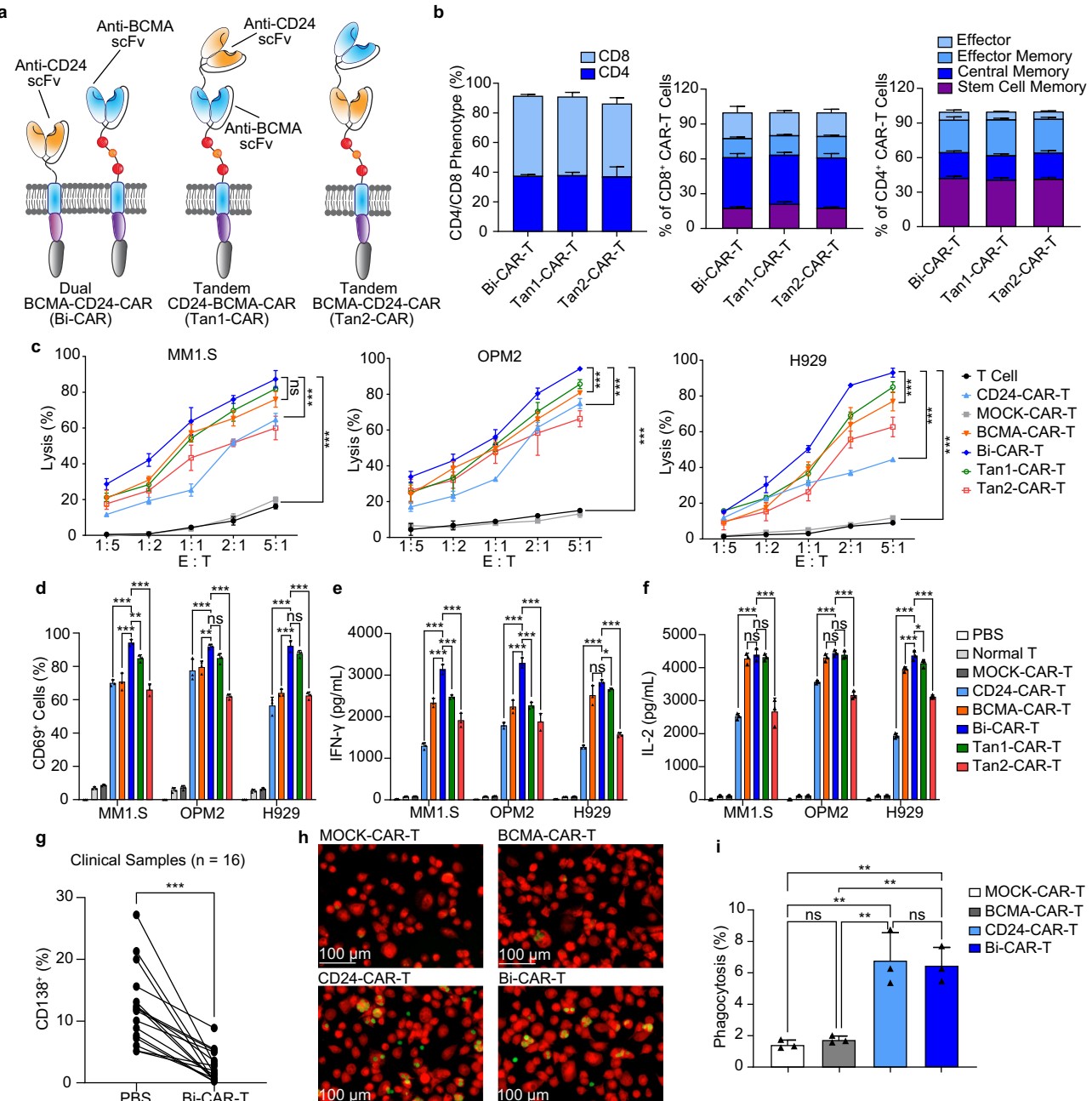

**Fig. 5 | Bispecific CAR-T cells had increased cell killing and macrophage phagocytic clearance in vitro. a** Dual BCMA-CD24-CAR (Bi-CAR), Tandem CD24-BCMA-CAR (Tan1-CAR), and Tandem BCMA-CD24-CAR (Tan2-CAR) constructs. The dual BCMA-CD24-CAR vector with two complete CAR units: BCMA-CAR and CD24-CAR. P2A was inserted between these 2 CAR vectors. Two tandem CAR vectors were constructed by CD24-scFv and BCMA-scFv, and two scFvs were linked with a (G4S)4 linker. The safety switch RQR8 was integrated into the hinge regions. **b** The phenotype of CAR-T cells, including the ratio of CD4 and CD8 phenotypes and the ratio of different memory phenotypes (*n* = 3 independent experiments). **c** Bispecific CAR-T cells cytolytic activity in vitro. CAR-T or T cells were added at the effector/target (E/T) ratio from 1:5 to 5:1. After 24 h of coculture, cytolytic activity was measured (*n* = 3 independent experiments). **d** The expression of T cell activation marker CD69 (*n* = 3 independent experiments). **e** IFN-γ concentrations on supernatants (*n* = 3 independent experiments). **f** IL-2 concentrations on supernatants

(*n* = 3 independent experiments). **g** Patient samples with Bi-CAR-T cells or PBS treatment for 24 h (*n* = 16). The percentage of the subpopulation of CD138⁺ cells decreased in 16 of 16 primary myeloma samples post-CART treatment. **h** Phagocytosis was performed by coculture of OPM2 cells that expressed GFP (green), DiD-stained macrophages (red), and CAR-T cells at a ratio of 2:1:1. After a 4-h coculture, suspended cells were washed and detected. Fluorescent images of phagocytic clearance (*n* = 3 independent experiments). The experiment was repeated twice with the same results. **i** Bar plot showing the percentage of phagocytosis detected by flow cytometry analysis (*n* = 3 independent experiments). Data are presented as mean values ± SD in (**b**–**f**) and (**i**). One-way ANOVA was used in (**c**), (**d**), (**e**), (**f**), (**i**) and paired t-test was used in (**g**). All tests are two-sided. *$P < 0.05$, **$P < 0.01$, ***$P < 0.001$, ns = $P > 0.05$. Raw data are provided in the Source Data file. Exact *P* values for each comparison shown in (**c**, **d**) and (**f**, **g**) can be found in Supplementary Data 1.

responses is limited and even patients with initial complete responses relapse[1,5,13]. Relapses may be due to several factors, including loss of BCMA expression or minimal residual MM cells that are resistant to this therapy[14–16]. To mitigate these limitations, combining BCMA-CAR T

cells with different targets and bispecific CAR-T cells have been introduced[36,37]. The CD19 and BCMA-CAR-T cell combination therapy has induced durable responses in patients with RRMM[38,39]. Although expression of CD19 is rare in plasma cells, there is evidence of the

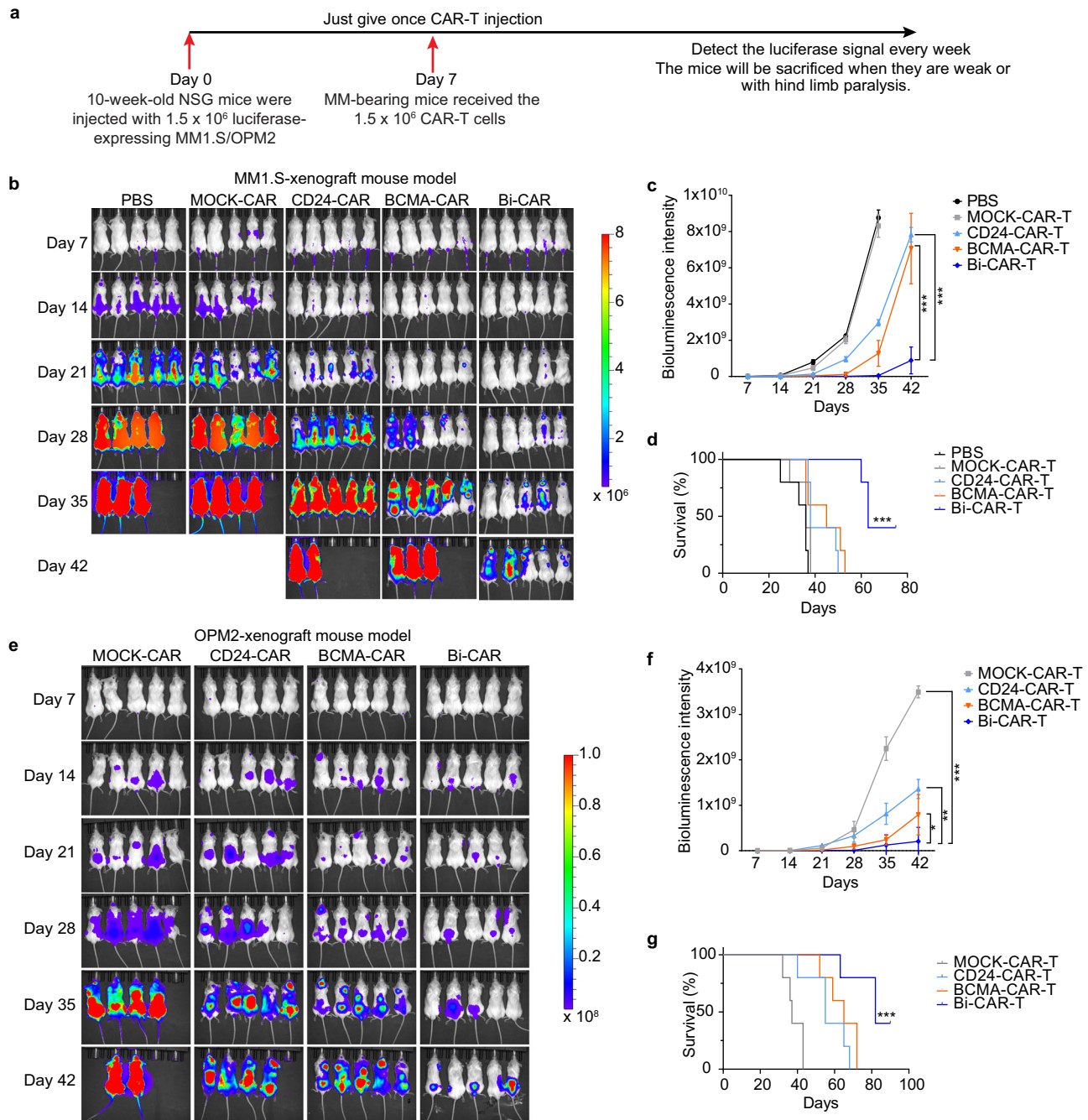

**Fig. 6 | Bispecific CAR-T cells eliminate MM cells in vivo. a** Schematic of the experimental setup. NSG mice were administered $1.5 \times 10^6$ MM1.S or OPM2 cells by intravenous injection. On day 7 after MM-cell injection, $1.5 \times 10^6$ CAR-T cells were administered. Mice were weighed and monitored for signs of distress every 3 days. Myeloma progression was monitored every 7 days until the mice developed hind-limb paralysis ($n = 5$ mice per group). **b** Bioluminescence images of MM1.S MM-bearing mice treated with CAR-T cells or PBS ($n = 5$ mice per group). **c** Quantitative analysis of bioluminescence signals of (**b**). Data are presented as mean values ± SD.

**d** Kaplan-Meier survival analysis of the experiment shown in (**b**). **e** Representative bioluminescence images of OPM2 MM-bearing mice treated with CAR-T cells or PBS ($n = 5$ mice per group). **f** Quantitative analysis of bioluminescence signals of (**e**). Data are presented as mean values ± SD. **g** Kaplan-Meier survival analysis of the experiment shown in (**e**). One-way ANOVA was used for statistical analysis. ${}^{**}P < 0.01$, ${}^{***}P < 0.001$, ns = $P > 0.05$. Raw data are provided in the Source Data file. Exact $P$ values for each comparison shown in (**c, d**) and (**f, g**) can be found in Supplementary Data 1.

---

existence of a small population of CD19+ MM cells that have drug-resistant, disease-propagating properties. The presence of which has been associated with high-risk disease, poor prognosis, early relapse, and reduced survival[40,41].

Our previous studies have found that CD24 is a marker of minimal residual disease in MM[17]. Normally a lymphoid-myeloid restricted gene product, CD24 is highly expressed in the light chain-restricted side-population cells of primary human MM. High-dose alkylating

chemotherapy treatment failures in MM are most likely due to the persistence of slow or non-cycling MM cells. CD24+ MM cells show minimal residual disease features with increased clonogenic potential and drug resistance. Furthermore, it was observed that more myeloma cells expressed more frequently CD24 upon relapse subsequent to BCMA CAR-T cell therapy[22].

Anti-CD24-based treatments have been evaluated in preclinical studies. CD24 monoclonal antibodies have been used in various solid

tumor models[42–45]. SWA11 is a high-affinity anti-CD24 antibody with strong cytotoxic activity and low off-target toxicity[42,43]. To date, there is no report of SWA11 being used in clinical trials for multiple myeloma. However, research teams have demonstrated the favorable effects of SWA11 in preclinical studies. Specifically, the combination of SWA11 with bortezomib (BTZ) resulted in an extension of the survival of multiple myeloma mice[17]. Additionally, Schambach et al. developed SWA11-based anti-CD24 CAR-NK cells with cytotoxicity against ovarian cancer cell lines and patient-derived ovarian cancer cells[29].

In this study, we verified that CD24-positive MM cells were indeed increased in residual MM cells after BCMA CAR-T-cell therapy. We developed CD24-CAR-T cells by selecting the scFv from SWA11 for the CAR against CD24. CD24-CAR-T cells efficiently decreased dormant (GFP+DiDHi) MM cells in the murine 5TGM1 MM model. Simultaneously, CD24-CAR-T cells blocked the CD24-Siglec-10 pathway and promoted phagocytic clearance by macrophages. Inhibition of the CD24-Siglec-10 pathway leads to the activation of inflammatory signaling and antitumor signaling. Additionally, we found that M2-like type macrophages were significantly increased in mice with 5TGM1, but significantly decreased in mice with 5TGM1 treated with CD24-CAR-T. MM cells produce chemokines such as C-X-C motif chemokine 12 (CXCL12) and C-C chemokine ligand 2 (CCL2). CD24+ MM cells can polarize C-X-C chemokine receptor type 4 (CXCR4)-positive macrophages towards the M2-like phenotype[46–48]. This phenomenon was reversed after CD24-CAR-T cell treatment. As MM cells were eliminated by CD24-CAR-T cells, the level of CXCL2 decreased. IFN-γ and TNFα released by activated CD24-CAR-T cells promoted macrophage polarization toward the M1-like lineage[49,50]. This reverses the immunosuppression caused by excessive M2-like macrophages in the MM microenvironment. Our data demonstrated that CD24-CAR-T polarizes macrophage to a M1-like phenotype supporting that CD24-CAR-T may modulate macrophage-related immune surveillance.

To further improve antitumor efficacy, bispecific BCMA-CD24-CARs were constructed. To find optimal CAR signaling, we designed 3 types of bispecific CAR-T cells. Bi-CAR-T cells displayed superior activity in vitro and in vivo against MM cells. Bioluminescence imaging showed nearly complete tumor clearance in MM xenograft mouse models. Additionally, mice treated with Bi-CAR-T cells exhibited significantly increased survival compared to control mice in the xenograft mouse mode.

In our in vitro cytotoxicity experiments, we observed an interesting phenomenon. At the ratio 5: 1 of CD24-CAR-T versus MM cells, the percentage of lysed MM cells were 78.9%, 65.0%, and 38.7% in the OPM2, MM1.S, and H929 cell lines, respectively (Fig. 2c). The CD24+ proportions for the three cell lines were: OPM2 (28.3%), MM1.S (11.7%), and H929 (7.77%) (Supplementary Fig. 4). The actual MM lysis rate is greater than the proportion of CD24+ cells, suggesting that CD24- cells are also being killed. This could be due to bystander effects of CAR-T cells. The term bystander effect refers to the phenomenon where CAR T-cells not only target cancer cells expressing a specific antigen but also stimulate cytotoxic activity against nearby cancer cells that lack the targeted antigen expression[51,52]. Because our in vitro cytotoxicity assays were conducted in 96-well plates, when CAR-T cells targeted CD24+ MM, CAR-T cells were activated and released large number of cytokines (such as IFN-γ, IL−2, TNF-α, perforin, and granzyme) to kill MM cells.

Another interesting phenomenon was that in our in vitro experiments, there was not much difference between Bi-CAR-T and single-CAR-T (especially BCMA-CAR-T) (Fig. 5c). However, in the in vivo experiments, Bi-CAR-T demonstrated a significantly more effective tumor killing effect (Fig. 6). The better in vitro inhibition of MM cells could be due to bystander effects. The BCMA- or CD24- MM cells were also killed by these cytokines released by CAR-T cells. However, in the multiple myeloma mouse models, the overall environment is significantly different from in vitro conditions. Unlike solid tumors that

are located in specific areas, multiple myeloma cells grow in large clusters, but are also distributed throughout the bone marrow, which may potentially reduce the impact of bystander effects. On the other hand, the escape of BCMA- or CD24- MM cells after the single CAR-T treatment were amplified over time in vivo. In the early stages of single CAR-T treatment in MM1.S and OPM2-xenograft mouse model models, there was a good response (day 7 to day 21). However, myeloma relapsed in the later stages of single CAR-T treatment. Dual CAR, which targets both BCMA and CD24 simultaneously, should provide better inhibition of tumor relapse.

In CAR-T cell treatment, safety is one of the major challenges, including precise tumor targeting to avoid off-target or on-target/off-tumor toxicity[53,54]. CD24 is expressed on various hematopoietic cells, such as B cells[42] and eosinophils[55], and is also found in non-hematopoietic cells, including neural cells[56], epithelial cells[57], pancreatic cells[58], and some cancer cell types[59]. This inevitably raises concerns about the safety of CD24-CAR-T therapy. After all, neurotoxicity has always been an issue that needs to be addressed and prevented in CAR-T treatments[60]. To promptly terminate the function of CAR-T in case of adverse reactions, we integrated RQR8, a suicide molecule, as an immunological safety switch into our design[30]. We have not found safety problems in our in vivo studies and we are conducting further research to avoid off-target toxicity. Meanwhile, to eliminate off-target effects at the source, we are constructing a CAR-T that will permit activation of CD24-targeted CARTs only in a MM microenvironment. This inducible CAR-T design relies on an autocleavable receptor construct (synNotch)[61] that activates CD24-targeted CAR expression upon recognition of MM specific makers (e.g. CD38, SLAMF7) on the surface of myeloma cells.

During the analysis of single-cell sequencing data, we identified an intriguing cell population. As shown in the first panel of Supplementary Fig. 1, a small subset of plasma/MM cells expresses CD3D. which we and others have observed similar expression patterns in MM plasma cells and Waldenström's macroglobulinemia (WM) B cells[62,63]. Currently, it is not clear why these B and plasma cells express T cell marker. One potential possibility is tumor cell dedifferentiation. Given that CAR-T therapies are predominantly manufactured using CD3+ T cells, the fate of this cell subset in the CAR production process warrants further investigation in subsequent studies.

This work presents a rational approach to engineering BCMA-CD24-CAR-T cells that not only effectively target bulk MM tumor cells, but also substantially reduce the numbers of minimal residual MM cells and modulate the tumor microenvironment (Fig. 7). This work supports the potential of BCMA-CD24-CAR-T cells as a multi-modality immunotherapy treatment approach for MM.

## Methods

### Study design

This study aimed to develop CD24-CAR-T cells and BCMA-CD24-CAR-T cells targeting multiple myeloma. To assess CAR efficacy, both in vitro and in vivo functional assays were conducted using various multiple myeloma tumor cell lines. An Institutional Review Board (IRB) approved the acquisition of anonymous human healthy donor PBMC products and diagnosed MGUS, SMM, or MM patients' bone marrow samples from University of Arkansas for Medical Sciences Myeloma Center Tissue Biorepository and Procurement Core (protocols IRB 261817 and 261821). The study included a total of 126 participants, comprising 56 females and 70 males, with 33 individuals identifying as African Americans and 93 as Caucasians. The average age was 62 ± 8.8 (mean ± SD). Sex and race were not used as a variable in study analyses, and study findings are not specific to one sex or race. All participants provided written informed consent for sample procurement in accordance with the Declaration of Helsinki. All the samples were de-identified. All participants provided informed consent and did not receive any compensation. The recruitment process did not involve

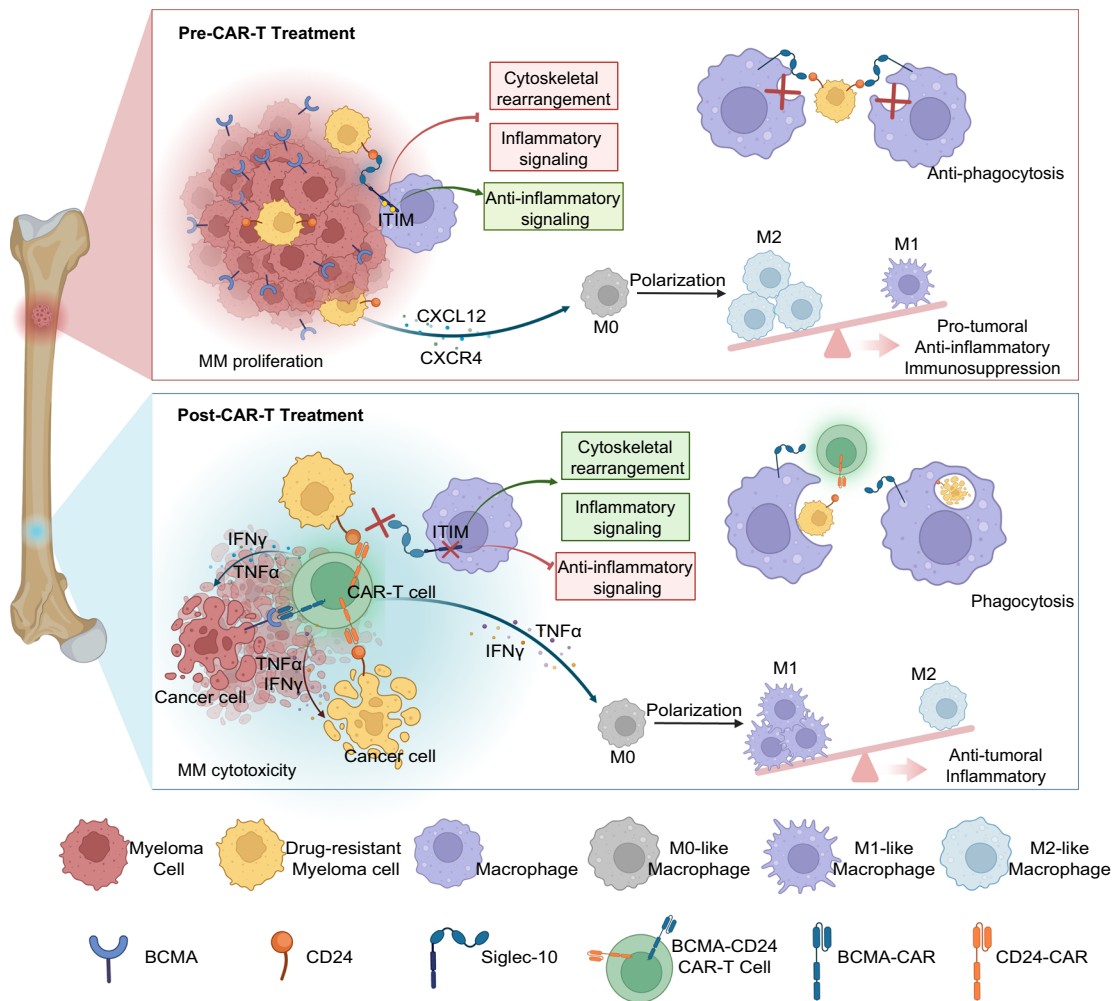

**Fig. 7 | Schematic representation of the roles of the BCMA-CD24-CAR-T cells in MM microenvironment.** MM cells expressing CD24 exhibit features of minimal residual disease. In MM microenvironment, CD24 is an important checkpoint molecule for controlling the innate immune response. When CD24 on tumor cells combines with Siglec-10 on macrophages, it causes immune receptor tyrosine inhibitory motif (ITIM) region to be phosphorylated, thus blocking Toll-like receptor-mediated inflammation and activating a series of intracellular signal pathways to achieve effective immunosuppression, promoting tumor immune escape and inhibiting cytoskeletal rearrangement, which blocks the macrophage phagocytic clearance. Additionally, MM cells produce chemokines, such as CXCR4 and CXCL12 that promote macrophage migration to the tumor niche and polarize macrophages toward an M2-like phenotype. We developed BCMA-CD24 CAR-T cells that were activated in vitro and in vivo against both bulk MM cells and minimal residual MM cells. Meanwhile, BCMA-CD24 CAR-T cells could block the CD24-Siglec-10 pathway and promote macrophage phagocytic clearance. Inhibition of the CD24-Siglec-10 pathway also leads to the activation of inflammatory signaling and antitumoral signaling. The IFN-γ and TNFα released by activated BCMA-CD24 CAR-T cells can also promote macrophages polarized to M1-like macrophage phenotype and reverse the immunosuppression caused by excessive M2-like macrophage phenotype in the MM microenvironment.

any selection bias. All animal protocols undergo a strict approval process with the Institutional Animal Care and Use Committee (IACUC) at University of Arkansas for Medical Sciences. The protocol number is IACUC 3997.

**Patients single-cell RNA sequencing (scRNA-seq) data process**
The scRNA-seq data of myeloma patients pre- and post-BCMA-CAR-T infusion were obtained from Gene Expression Omnibus GSE210079 database. We collected five pre-infusion samples (01-pre, 16-pre, 19-pre, 32-pre, 33-pre) and six at day 28 post-infusion samples (01-d28, 16-d28, 19-d28, 20-d28, 27-d28, 33-d28,) with clinical information. The median progression-free survival (PFS) is 190 days (range 57–1053 days, $n = 7$).

The raw scRNA-seq data were loaded into R through the Seurat V4 package. The Seurat's integration workflows were used for quality control. First, cells exhibiting low-complexity libraries, indicating detection of transcripts aligned to fewer than 200 genes, potentially representing dying or apoptotic cells (more than 10% of unique molecular identifiers stemming from mitochondrial genes), as well as cells with high-complexity libraries (with detected transcripts aligned to more than 7000 genes), were excluded. We used DoubletFinder function to identify potential doublets within our dataset. Essentially, a doublet is characterized as a single-cell library that represents more than one cell. Upon closer inspection of certain known markers, it was observed that the implicated cluster comprises doublets of more than one cell type, as no cell type is recognized for robustly expressing both markers simultaneously. Doublets were individually removed from each sample, employing an anticipated doublet rate of 0.05. Subsequently, the cell expression matrix of each sample underwent normalization using the NormalizeData function with default parameters. Following this, the FindVariableFeatures function, with default parameters, was employed to identify highly variable genes (HVGs) within each normalized matrix. The SelectIntegrationFeatures function was then utilized, specifying nfeatures = 2000, to select genes for the integration of multiple samples. To mitigate the impact of the cell cycle on data integration, we excluded cell-cycle-related genes from the gene set. Sequentially, the RunPCA and ScaleData functions were

applied with the parameter features set to these selected genes. Subsequent to scaling the matrix for each dataset, PCA was conducted. To reduce batch effects, we applied the "anchor" integration method (functions FindIntegrationAnchors and IntegrateData)[64]. We employed the FindIntegrationAnchors function, specifying reduction = 'rpca,' to identify a set of anchors between all matrices. These anchors were used to integrate the matrices through the IntegrateData function with parameter dims = 1:50. Finally, the ScaleData function was applied to scale the integrated matrix using default parameters. Then the nitration results were used as input for clustering with the Louvain algorithm with multilevel refinement and UMAP. The gene-specific markers for each cluster were determined using the FindMarkersAll function with MAST test statistics. Subsequently, the top 20 gene-specific markers were input into the CellMarker 2.0 cell annotation tool[65] to obtain automatic annotations. Then, cluster annotations were generated through a combination of automatic annotation and manual annotation based on relevant studies[66,67]. Gene-set enrichment analysis of these marker genes was carried out for kyoto encyclopedia of genes and genomes (KEGG) pathway analysis. Differentially expressed genes (DEGs) were denoted as statistically significant for the false discovery rate (FDR) less than 0.05 with a fold change exceeding 1.2. All original code has been deposited to GitHub (DOI: 10.5281/zenodo. 10014735).

## Patient samples and cell lines

PBMCs and BMMCs were obtained from University of Arkansas for Medical Sciences (UAMS) Myeloma Center Tissue Biorepository and Procurement Core. The UAMS institutional review board approved these research studies (IRB 261817 and 261821). Human MM1.S (ATCC#: CRL−2974), RPMI 8226 (ATCC#: CCL-155) cells were purchased from ATCC. Mouse 5TGM1 cells were purchased from Harlan Laboratory Inc (Reference#: 799). Human OPM2, MM1-144, KMS-34, FR4, XG-7, MMM1, KAS-6/1, H929, Delta 47, OCI-MY5, AMO1, JIM3, VPC-6, KMS12-PE, ARP-1, KMS-11, KMS12BM, JK-6L cells were generously provided by Dr. Siegfried Janz (Medical College of Wisconsin, Milwaukee, WI, USA). All cell lines' authenticity was confirmed through STR (Short Tandem Repeat) profiling, and routine mycoplasma testing was performed using a mycoplasma detection kit. These cell lines were cultured in Roswell Park Memorial Institute (RPMI) 1640 medium containing 10% fetal calf serum (FBS) (Gibco), 100 U/ml penicillin, and 100 µg/ml streptomycin (Gibco).

## Construction of CAR vector

The human CD24-CAR vector was constructed containing CD24-specific scFvs (derived from SWA11)[28,29], a safety switch (RQR8)[30] in the hinge region, and a human 4-1BB co-activation domain with human CD3ζ. The human MOCK-CAR vector only contains a safety switch in the hinge region and a 4-1BB co-activation domain with CD3ζ (Fig. 2a). The mouse CD24-CAR vector was constructed containing CD24-specific scFvs, a RQR8 and a mouse CD28 co-activation domain with mouse CD3ζ. The mouse MOCK-CAR vector only contains a safety switch and a mouse CD28 co-activation domain with mouse CD3ζ. The human BCMA-CAR vector contains BCMA-scFv (derived from clone C11D5.3)[36,68], a safety switch in the hinge region, and a 4-1BB co-activation domain with CD3ζ.

The dual-targeted BCMA-CD24 CAR vector with two complete CAR units: BCMA-CAR and CD24-CAR. A P2A self-cleaving peptide was inserted between the 2 CAR vectors. Two tandem CAR vectors were constructed by linking CD24-scFv and BCMA-scFv with poly-Glycine-Serine (G4S)4 peptide linker (Fig. 5a). To decrease the risk of severe immunological side effects, we integrated RQR8, an immunological safety switch, with a CD34 epitope and 2 CD20 mimotopes as a suicide molecule. All vectors contain SFFV promoters to drive the expression of CARs.

## Transduction of T cells

Density-gradient centrifugation was used to isolate PBMCs from three healthy donor samples by Ficoll-Paque (General Electric). Human T cells were isolated from PBMCs with human CD3 Microbeads (Miltenyi) and were cultured in AIM V Medium (Thermo Fisher Scientific) with 5% human AB serum (Sigma-Aldrich) and 400 IU IL−2 (R&D Systems). Dynabeads human T-activator CD3/CD28 (Thermo Fisher Scientific) was added for human T-cell expansion and activation[31]. T cells were activated for 2−3 days prior to transduction. Mouse T cells were isolated from the spleens of eight-week-old C57BL/KaLwRij mice with mouse CD3ε Microbeads (Miltenyi) and were cultured in AIM V Medium (Thermo Fisher Scientific) with 10% FBS (Gibco) and 400 IU IL-2 (R&D Systems). Dynabeads mouse T-activator CD3/CD28 (Thermo Fisher Scientific) were added for mouse T-cell expansion and activation. T cells were activated for 2−3 days prior to transduction.

Lentivirus particles were used to transduce human T cells or mouse T cells. T cells and concentrated lentivirus were added into RetroNectin precoated plates (Takara Bio)[31]. Cells were cultured in AIM V Medium for 24 h, and the transduction step was repeated. After 24 h, cells were washed with PBS and cultured in the fresh medium for 7 days. CAR-T cells were detected by BD FACSVerse™ flow cytometry with CD34 antibodies (Invitrogen, Cat#: MA1-10205, Clone:QBEND/10) and CD34 Microbeads (Miltenyi) for isolation. Cell Counting Kit-8 (APExBIO) was used to detect CAR-T-cell proliferation. Phenotypes of CAR-T cells were detected by BD FACSVerse™ flow cytometry. T cells memory phenotypes were assigned according to CD62L and CD45RO expression within the CAR-T cell population as follows: stem cell memory (CD45RO⁻/CD62L⁺), central memory (CD45RO⁺/CD62L⁺), effector memory (CD45RO⁺/CD62L⁻), effector cells (CD45RO⁻/CD62L⁻)[69]. Flowjo V10 (BD) and GraphPad Prism 9 (GraphPad Software Inc.) were used for flow cytometry analysis.

## CD24 knockout

The lentiCRISPR v2 plasmid was kindly provided by Feng Zhang (Addgene plasmid # 52961)[70]. To create the lentiCRISPR v2-CD24 plasmid, we synthesized two gRNA sequences targeting exon 1 and exon 2 of CD24, as follows: gRNA1: 5′-AGGGCCTCACCTGCGTGGGT-3′ and gRNA2: 5′-ATTTGGGGCCAACCCAGAGT-3′. These gRNA sequences were integrated into the lentiCRISPR v2 vector using BsmBI restriction sites. Subsequently, the lentiCRISPR v2-CD24 plasmids, along with psPAX2 and pVSV-G plasmids, were co-transfected into HEK293T cells using Lipofectamine2000 (Invitrogen). Lentivirus was collected at transduced MM cells. The transduced cells were selectively cultured using 2.5 µg/ml puromycin (Sigma, USA). Following one week of puromycin selective cultivation, the selected myeloma cells were individually cloned using limited dilution. The knockout of CD24 was confirmed by BD FACSVerse™ flow cytometry.

## Cytotoxicity assay

We detected the cytolytic activity of CAR-T cells based on fluorescence intensity. In 96-well plates, $20 \times 10^3$ mCherry-positive MM cells were seeded. CAR-T cells or T cells were added at the effector/target (E/T) ratio from 1:5 to 5:1. Then fluorescence intensity was measured at excitation 585 nm/emission 620 nm with a plate reader (BioTek Cytation 5 Cell Imaging Multimode Reader) after 24 h of coculture. GraphPad Prism 9 (GraphPad Software Inc.) were used for fluorescence-based killing assays.

$$\%lysis = \frac{experimental\ lysis - spontaneous\ lysis}{maximal\ lysis - spontaneous\ lysis} \times 100\%$$

We also detected the T-cell activation marker CD69 and measured the cytokines IFN-γ, IL-2, and TNF-α, which are produced by activated CAR-T cells, in the supernatant after 24 h of coculture.

## Macrophage phagocytosis

BMMCs were cultured in Iscove's Modified Dulbecco's Media (IMDM) medium with 10% human AB serum and macrophage colony-stimulating

factor for 7 days to induce macrophage differentiation. IFN-γ and lipopolysaccharide (LPS) were added on day 3. The mature macrophages were stained with DiD dye before phagocytotic experiments. In vitro phagocytosis was performed by coculture of OPM2, MM1.S, or H929 cells that expressed GFP, DiD stained macrophages, and CAR-T cells at a ratio of 2:1:1 for 4 h in the 37 °C incubator. Next, cells were washed with PBS twice, and GFP⁺DiD⁺ cells were detected by BD FACSVerse™ flow cytometry and ZEISS Axio Observer fluorescence microscope. Flowjo V10 (BD) and GraphPad Prism 9 (GraphPad Software Inc.) were used for flow cytometry analysis and Fluorescence microscope software ZEN3.5 (ZEISS) and GraphPad Prism 9 (GraphPad Software Inc.) were used for fluorescence image assays.

### Specificity and cross-reactivity assay

To verify the specificity and cross-reactivity of the CD24-CAR-T cells, flow cytometry and fluorescent imaging have been employed. The FITC-labeled human CD24, mouse CD24 and human BCMA (AcroBiosystems, #CD4-H52H3, #CD4-M52H7 and #CD4-M52H7) was generated using FluoroTag™ FITC Conjugation Kit (Sigma-Aldrich, #FITC1). Solute FITC and proteins were mixed at the molar ratio of 5:1 and incubated for 2 h at room temperature in a reaction vial with gentle stirring at dark room. Then, the conjugated product was purified by ultrafiltration to remove the unconjugated FITC molecule. $2 \times 10^5$ of CD24-CAR-T cells were stained with 100 μL of 3 μg/mL of FITC-labeled proteins and anti-human CD3-APC-CY7 antibodies (Biolegend, #300318). FITC and APC-CY7 signal were tested by BD FACSVerse™ flow cytometry and ZEISS Axio Observer fluorescence microscope. FITC-signal used to evaluate the binding activity of CD24-CAR-T cells.

### 5TGM1/KaLwRij MM mouse models

Eight-week-old C57BL/KaLwRij mice were purchased from the Harlan Laboratory (https://www.envigo.com/model/c57bl-kalwrijhsd). Mice used in the experiments are sex mixed. Mice were housed under social conditions (2–5 mice per cage) on a standard 12-h dark/12-h light cycle, ambient temperature 21 °C ± 1 °C, and humidity 50% ± 10%. All mice were housed in a pathogen-free animal facility with standard food and water. On day 0, mice were injected intravenously via the tail vein with either 100 μL PBS or $1 \times 10^6$ 5TGM1-GFP cells stained with DiD dye and randomized into three groups ($n = 5$ / group). Seven days after the 5TGM1 injections, mice were injected intravenously via the tail vein with either PBS (PBS group), $1 \times 10^6$ MOCK-CAR-T cells (MOCK-CAR T group), or CD24-CAR-T cells (CD24-CAR T group). Twenty-one days after the 5TGM1 injections, the experiment was terminated. All animal protocols undergo a strict approval process with the Institutional Animal Care and Use Committee (IACUC) at University of Arkansas for Medical Sciences. The protocol number is IACUC 3997. We determined tumor burden and humane endpoints based on the following criteria: euthanasia was performed on mice when signs of distress were observed, such as abdominal distension due to peritoneal ascites, difficulty or labored breathing, significant weight loss, impaired mobility, or signs of paraplegia, using carbon dioxide.

Serum protein electrophoresis (SPE) was performed by QuickGel Electrophoresis (Helena Laboratories). BMMCs were isolated by Ficoll-Paque. Dormant (GFP⁺DiD^Hi) and activated (GFP⁺DiD^Neg) cells were detected by flow cytometry[21]. The gating strategy for flow cytometry is shown in Supplementary Fig. 18. BM microenvironmental cells were sorted out with fluorescence-activated cell sorting by depleting 5TGM1-GFP⁺ MM cells and human CD34⁺ CAR-T cells. Sorted cells with a purity greater than 98% were used for scRNA-seq.

### Mouse single-cell RNA sequencing

Single-cell emulsions were generated with the Chromium Next GEM Chip G Single Cell Kit (10X Genomics) and the Chromium Next GEM Single Cell 3′ v3.1 Kit (10X Genomics) following the standard protocol. Libraries were assessed for mass concentration with the Qubit 1X

dsDNA High Sensitivity Assay Kit (Thermo Fisher Scientific). Library fragment size was assessed with the High Sensitivity NGS Fragment Analysis Kit (Agilent) on the Fragment Analyzer System (Agilent). Libraries were functionally validated with the KAPA Library Quantification Kit (Roche). Initial low-pass "surveillance" sequencing was performed on a NovaSeq SP 100-cycle Flow Cell (Illumina) and data were assessed with the Cell Ranger count (10X Genomics) output.

### Bioinformatic analysis of scRNA-seq

The raw scRNA-seq data were preprocessed with Cell Ranger v6 (10X Genomics) and reference genome of Mus musculus version mm10 to demultiplex for cell and transcript and generate count table. The count table was loaded into R through the Seurat V4 package for further analysis. The cells that have fewer than 200 genes, greater than 7000 genes, and more than 10% of unique molecular identifiers stemming from mitochondrial genes were discarded from the analysis. For individual samples, principal component analysis (PCA) was then performed on significantly variable genes of the remaining high-quality cells. The results of the individual samples were used for data integration across samples with the reciprocal PCA method to minimize technical differences between samples. Then the nitration results were used as input for clustering with the Louvain algorithm with multilevel refinement and UMAP. The gene-specific markers for each cluster were determined using the FindMarkersAll function with Model-based Analysis of Single-cell Transcriptomics (MAST) test statistics. Subsequently, the top 20 gene-specific markers were input into the CellMarker 2.0 cell annotation tool[65] to obtain automatic annotations. Then, cluster annotations were generated through a combination of automatic annotation and manual annotation based on relevant studies[66,67]. Gene-set enrichment analysis of these marker genes was carried out for KEGG pathway analysis.

### Western blotting for phosphorylation status of SHP-1/2

Mouse macrophages were isolated from BMMCs of 5TGM1/KaLwRij MM mouse models with CD11b MicroBeads (Miltenyi). Mouse macrophages were lysed in the mammalian cell extraction buffer (BioVision) with Protease and Phosphatase Inhibitors (ThermoFisher). Protein lysates were incubated on ice for 10 min and centrifuged at $14,000 \times g$ for 10 min at 4 °C. Proteins were separated with 4–12% Bis-Tris Gel (Invitrogen) at 120 V for 1.5 h. And then transferred to a nitrocellulose (NC) membrane for 1.5 h at 200 mA. The membrane was blocked for 1.5 h with 5% milk at room temperature. Primary antibodies SHP-1 (Cell Signaling Technology, catalog # 3759, clone # C14H6, 1:1000), SHP−2 (Cell Signaling Technology, catalog # 3397, clone # D50F2, 1:1000), Phospho-SHP-1 (Tyr564) (Cell Signaling Technology, catalog # 8849, clone # D11G5, 1:1000), Phospho-SHP-2 (Tyr580) (Cell Signaling Technology, catalog # 5431, clone # D66F10, 1:1000), β-Actin (Cell Signaling Technology, catalog # 4967, 1:1000), were incubated overnight at 4 °C. Anti-rabbit IgG, HRP-linked antibody (Cell Signaling Technology, catalog # 7074, 1:1000) were incubated for 2 h. ECL Substrates (Bio-Rad, catalog # 1705060) were used for exposure. Western Blotting imaging was exposed with a Bio-Rad ChemiDoc XRS⁺. The images were analyzed by Image Lab Software (Bio-Rad).

### Human MM cell lines xenograft mouse models

Eight-week-old NOD.Cg-Prkdc^scid Il2rg^tm1Wjl/SzJ (NSG) mice were purchased from the Jackson Laboratory (https://www.jax.org/strain/005557). Mice used in the experiments are sex mixed. Mice were housed under social conditions (2–5 mice per cage) on a standard 12-h dark/12-h light cycle, ambient temperature 21 °C ± 1 °C, and humidity 50 ± 10%. All mice were housed in a pathogen-free animal facility with standard food and water. On day 0, mice were administered $1.5 \times 10^6$ MM1.S or OPM2 cells by intravenous injection and randomized into 5 groups (n = 5/group). On day 7 after MM1.S or OPM2 cell injection, $1.5 \times 10^6$ CAR-T cells or PBS was administered. Mice were weighed and

monitored for signs of distress every 3 days. D-luciferin was intraperitoneally injected with 150 mg/kg, and bioluminescence images were acquired 10 min later using IVIS living image version 4.4 software (Caliper Life Sciences). All animal protocols undergo a strict approval process with the Institutional Animal Care and Use Committee (IACUC) at University of Arkansas for Medical Sciences. The protocol number is IACUC 3997. Myeloma progression was monitored by bioluminescence images every 7 days. We determined tumor burden and humane endpoints based on the following criteria: euthanasia was performed on mice when signs of distress were observed, such as abdominal distension due to peritoneal ascites, difficulty or labored breathing, significant weight loss, impaired mobility, signs of paraplegia, or the bioluminescence signal was more than $2 \times 10^{10}$ using carbon dioxide.

### Flow cytometry

Cells were washed with flow cytometry buffer (2% FBS in PBS) and then incubated with antibodies in the dark for 30 min on ice. Cells were then washed twice and 7-AAD was used to separate live/dead staining before analysis on a BD FACSVerse™. The following antibodies were used: APC anti-human CD24 Antibody (Biolegend Cat#: 311118, Clone:ML5, 1:200 dilution); APC/Cyanine7 anti-human CD24 Antibody (Biolegend Cat#: 311131, Clone:ML5, 1:200 dilution); FITC anti-human CD138 (Syndecan-1) Antibody (Biolegend Cat#: 352304, Clone:DL-101, 1:200 dilution); PE anti-human CD4 Antibody (Biolegend Cat#: 357404, Clone:A161A1, 1:200 dilution); APC anti-human CD8 Antibody (Biolegend Cat#: 344722, Clone:SK1, 1:200 dilution); APC/Cyanine7 anti-human CD45RO Antibody (Biolegend Cat#: 304227, Clone:UCHL1, 1:200 dilution); FITC anti-human CD62L Antibody (Biolegend Cat#: 304838, Clone:DREG-56, 1:200 dilution); APC/Cyanine7 anti-human CD3 Antibody (Biolegend Cat#: 300318, Clone:HIT3a, 1:200 dilution); PE anti-human CD34 Antibody (Invitrogen Cat#: MA1-10205, Clone:QBEND/10, 1:200 dilution); FITC Anti-Human CD69 Antibody (BD Cat#: 555530, Clone:FN50, 1:200 dilution); Anti-human CD24 Antibody (InVivo BioTech Cat#: AK208.2/06B.1, Clone: SWA11, 1:200 dilution); BV510 anti-mouse/human CD11b Antibody (Biolegend Cat#: 101263, Clone: M1/70, 1:200 dilution); BV605 Rat Anti-Mouse F4/80 Antibody (BD Cat#: 743281, Clone:T45-2342, 1:200 dilution); PE anti-mouse CD86 Antibody (Biolegend Cat#: 105008, Clone:GL-1, 1:200 dilution); PE/Cyanine7 anti-mouse CD206 (MMR) Antibody (Biolegend Cat#: 141720, Clone:C068C2, 1:200 dilution); PE Rat Anti-Mouse CD24 Antibody (BD Cat#: 553262, Clone:M1/69, 1:200 dilution). All data were collected using the BD FACSDiva v7 software and analyzed using FlowJo V10 (BD) Software.

### Statistical analysis

All data were analyzed with Prism 9 (GraphPad) and presented as means ± standard deviation (SD) unless otherwise indicated. For statistical analyses, a two-tailed Student $t$ test was performed for two group comparisons. One-way ANOVA analysis of variance was used to determine the statistically significant difference for multiple group comparisons. A $P$ value of 0.05 or less was considered significant.

### Reporting summary

Further information on research design is available in the Nature Portfolio Reporting Summary linked to this article.

## Data availability

The mouse single-cell RNA sequencing have been deposited in the Gene Expression Omnibus (GEO) database under accession code GSE226956. The human single-cell RNA sequencing publicly available data used in this study are available in the GEO database under accession code GSE210079. The remaining data are available within the Article, Supplementary Information, or Source Data file. Source data are provided with this paper.

## Code availability

No new algorithms were developed for this manuscript. All original code has been deposited to GitHub (https://doi.org/10.5281/zenodo.10014735).

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

## Acknowledgements

The study was supported by grants from the National Cancer Institute 1R01CA236814-01A1 (F.Z.), 3R01-CA236814-03S1 (F.Z.), and U54CA272691-01 (F.Z. and J.D.S.), U.S. Department of Defense CA180190 (F.Z., G.B) as well as funding from the Myeloma Crowd Research Initiative Award (F.Z.), the Paula and Rodger Riney Foundation (F.Z.), the Myeloma Solution Fund (F.Z., Q.Y., R.O.) and UAMS Winthrop P. Rockefeller Cancer Institute (WRCRI) Fund (F.Z.), and the Arkansas Breast Cancer Research Program (F.S.). We are indebted to the clinicians of the Myeloma Institute for Research and Therapy for referring patients to this study and to all the patients who have helped us in our pursuit of a cure. Figures 1a, 3a, and 7 were created with BioRender.com. We thank iCell Gene Therapeutics Inc. for constructing and providing the vectors of the CARs. We thank the UAMS Genomics Core for single-cell RNA sequencing and Tissue Biorepository and Procurement Service (TBAPS) for providing primary samples.

## Author contributions

F.S. and Y.C. performed the experiments, collected, analyzed the data, generated the figures, wrote, and edited the manuscript; V.W., W.G. analyzed the scRNA-seq data; D.M., H.X., D.G., E.S., C.B., C.A., S.H., C.S., S.T., Y.M., Q.Y., R.O., M.Z., F.v.R., S.J., G.B., G.T. reviewed the data and provided guidance for revision; G.T. and J.D.S. analyzed and interpreted data and revised the manuscript. F.Z. conceptually developed this project, designed and supervised this study, collected and analyzed data, wrote and edited the manuscript. All authors discussed the results and commented on the manuscript.

## Competing interests

F.S. and F.Z. are inventors on the patent application "Bispecific BCMA-CD24-CAR-T design", describing the therapeutic use of BCMA-CD24-CAR-T cells for the treatment of multiple myeloma. The patent number is pending. The other authors declare no competing interests.
