## [Peer Review File · Nature Communications]

Bispecific BCMA/CD24 CAR-T Cells Control Multiple Myeloma GrowthREVIEWER COMMENTS

Reviewer #1 (Remarks to the Author): with expertise in CAR-T, multiple myeloma

This study presents data on novel BCNA-CD24 CART cells against multiple myeloma. This is good executed study. I have the following suggestion. Figure 6. Please add data on mouse weight in the study.

Reviewer #3 (Remarks to the Author):

I co-reviewed this manuscript with one of the reviewers who provided the listed reports as part of the Nature Communications initiative to facilitate training in peer review and appropriate recognition for co-reviewers.

Reviewer #4 (Remarks to the Author): with expertise in CAR-T, computational

General comment:

First I am delighted to serve as a reviewer for this work. In this work, Sun and Cheng et al developed CD24-CAR-T cells and analyzed in-vitro and in vivo data to assess the killing ability of MM cells. They further investigated the tumor microenvironment changes, particularly macrophage phenotypic alterations following CAR-T treatment. Finally, the authors constructed bispecific CAR-T cells, BCMA-

CD24-CAR-T. The subsequent functional analysis, which showcases enhanced efficacy compared to monospecific BCMA-CAR-T-cell therapy, holds immense clinical significance.

Overall the study demonstrated a large amount of work; nevertheless, there are several areas that require significant improvement. Primarily, the data analysis of the single-cell RNA sequencing (scRNAseq) segment appears to be incomplete and lacks the necessary depth to provide comprehensive insights. Additionally, the organization of the manuscript needs improvement, as it currently lacks a cohesive and well-connected structure.

My major concerns are stated as the following:

1. I observed CD3D expression in part of Plasma/MM clusters in the umap of supplementary figure 1, any chances that these are B and T cell doublets?
2. Related to figure 1e, a boxplot showing the expression level of CD24 of all patients' MM cells in D28 vs. Pre-infusion would be very nice to help the readers to get an overall expression difference.
3. It would be more robust if the authors could include additional public scRNA-seq cohorts showing the either increased expression of CD24 or increased expression frequency of CD24 in MM cells.
4. Related with figure 1h, is CD24 expression level also increased in MM after Bortezomib treatment? or the treatment only affected the frequency of CD24+ MMs?
5. In figure 2h and 2i, MOCK-CAR-T show similar phagocytosis level to PBS. Did the authors compared the phagocytosis level by using other MM cell lines with different CD24 expression %? For example, MM1.S or H929?
6. Regarding the single cell RNAseq analysis in figure4, did the author observed other celltypes except the lymphoid/myeloid cell populations shown in the figure?
7. Did the author performed cell cell communication analysis using CAR-T cells and macrophages captured in scRNAseq? What are the most involved communication signals in treatment group?
8. How did the authors perform celltype annotation? I couldn't find the detailed description in the method part. Did the author applied independent cell type annotation tools, including celltypist, singleR etc to validate the robustness of celltype annotation for scRNAseq results shown in figure 1b and in figure 4a&b? Relying solely on a single annotation result may introduce vulnerabilities and potential biases in the analysis.
9. The authors should evaluate batch effects by employing appropriate computational methods like ComBat, Harmony, or Seurat's integration workflows.

Minor comments:

1. Minor comment: line 245-246: I didn't see evidence for blocking of cd24/siglec-10 signaling in figure 4e. suggest to move downwards after introducing results of figure 4f,g and h.

Reviewer #5 (Remarks to the Author): with expertise in CAR-T

Sun and colleagues have submitted a manuscript, where they identified CD24 as a marker of relapsing multiple myeloma cells after anti-BCMA CAR therapy and hypothesized that CD24 may also play a role in immune evasion of MM in this context. They went on to design new CAR towards CD24 and dual CD24-BCMA CAR. They demonstrate activity in vitro and in vivo and hypothesise that CD24-CAR may be particularly acting through macrophages and show macrophage reprogramming in vivo. As such the concept is novel and innovative. The paper is well written and easy to follow. In the current status, I have concerns with several of the conclusions, which I do not think they are fully backed by data, as follows:

1) For their experiments, the authors claim to use cell lines with low expression of CD24 (range 7-28%). Still when performing killing experiment most cells are lysed, which raises concerns as to the specificity of the CAR. While the authors have used CD24 negative cells (HEK); in the light of this data MM CD24KO should be used.

2) The authors use headless CAR as controls, which may be adequate in certain settings but not in all (see 3)

3) A major problem I see in the experiments investigating macrophage phagocytosis in vitro and in vivo is the lack of a control CAR actually killing MM cells efficiently but not addressing macrophage function. One could easily imagine that dying cells by CAR killing could be more easily phagocytated. In fact other groups have implied macrophage reprogramming in CAR efficiency. This must be properly demonstrate to showcase the mechanistic advance here

4) Activity of dual CAR over single CAR is not overly convincing in vitro, while very convincing in vivo (which is also more relevant), this must be better contrasted in the text.

Formal aspects

1) How was cell line identity ascertained

2) It is difficult to follow in the figures the amount of replicates and independent repeats performed and where data was pooled. This must be made crystal clear for each and every figure and subfigure

Reviewer #2 (Remarks to the Author): with expertise in multiple myeloma

Title: Bispecific BCMA/CD24 CAR-T Cells Control Multiple Myeloma Growth

Review:

The present study by Sun, Cheng et al. delves into the role of the CD24-Siglec10 immunological synapse between multiple myeloma cells and macrophages, and its potential as a therapeutic target for patients. The researchers focused on CD24 as a target and demonstrated that after anti-BCMA CAR-T treatment, the progression of the disease might be linked to slowly proliferating CD24-positive myeloma cells. To explore this further, they developed innovative anti-CD24 and bispecific anti-CD24/anti-BCMA CAR-T cells, and the results from in-vitro and xenograft mouse models are highly promising. Moreover, the study highlights an interesting finding related to macrophages. They observed an increased phagocytosis of myeloma cells by macrophages and changes in macrophage subpopulations, indicating a shift from M2 to M1 macrophages and a reversal of immune checkpoint signaling mediated by the blocking of CD24 on the myeloma cells.

Overall, the manuscript is well-written, follows a clear and logical structure, and represents the first study exploring the use of CD24+ CAR-T cells as a treatment strategy in multiple myeloma. The presented results are encouraging and hold great potential for further evaluation in clinical trials.

To enhance the manuscript further, some revisions are suggested:

- In lines 107ff, it's common practice to perform FDR correction of p-values for DEG analyses of scRNA sequencing data. Consider revising Fig 1E and the respective methods part (lines 390f.) to reflect this.
- Lines 124ff. Regarding the increased percentage of CD24+ MM cells after 24h incubation with bortezomib, it may not be appropriate to draw conclusions about drug resistance in such a short timeframe. Additional investigations into CD24 expression upregulation and testing with other drugs (e.g., dexamethasone, lenalidomide) could provide more insights.
- Lines 245f. Please revise wording here: The authors have shown that CD24-CAR-T activate phagocytosis and lead to differences in the macrophage subpopulations. However, for a full demonstration of changes in the signaling cascade, a confirmation on protein level would be preferable, e.g., Western Blotting showing phosphorylation status of SHP-1/2/ITIM regions or by proteomics. Moreover, the reference to Figure 4e seems to be misplaced. Fig 4e only shows that macrophages are present in different proportions in the BM and is neither related to function nor CD24/Siglec10 pathway.
- Lines 248ff. and Fig 4f. What are the percentual changes between the macrophage clusters? Do the percentages differ? The authors state "Cluster 0 was increased in MM-bearing mice, but decreased to normal levels after CD24-CAR-T cells treatment." However, the total amount of cells captured after CD24-CAR-T is less compared to PBS and MOCK-CAR-T conditions making it difficult to read if the differences between conditions are also reproducible in relative numbers.
- In lines 251f, the statement "The differentially expressed genes were divided into upregulated genes and downregulated genes" is trivial and can be deleted.
- In lines 276ff, emphasis should be placed on comparing lysis rates between BI-CAR-T and single CD24-CAR-T cells, rather than comparing with MOCK-CAR-T, which is trivial and bears no new insights.
- Regarding the lysis capacity differences between cell lines (lines 276ff / Fig. 5), consider discussing the BCMA expression levels of these cell lines, as variations in BCMA expression may contribute to the differences observed.
- In line 325, correct the typo: change "ant-CD24" to "anti-CD24."

- Lines 324ff. Are there pre-clinical or clinical studies of SWA11 in the setting of multiple myeloma? If not, I would add this information to elevate the discussion.
- In lines 353ff, the statement "CD24 is rarely expressed in normal human tissues" contradicts data from the Human Protein Atlas. Here, CD24 has high protein expression scores among various tissues including the CNS, lungs, intestines and reproductive organs. The data regarding potential toxicity presented in the study is just restricted to liver, kidney and spleen. We would highly recommend to revise this statement and – if no further data is available to show otherwise – to include a limitations section discussing adverse events, that might be encountered. The high expression in brain is particularly worrisome, specifically in the context of CAR-T-mediated neurotoxicity.

Appended below are the reviewers' comments. Our point-by-point responses have been inserted, including descriptions of changes made in the amended manuscript. The reviewers' comments are indicated in blue, while our responses are indicated in black. The revised parts in the manuscript are highlighted in yellow.

Reviewer #1

This study presents data on novel BCMA-CD24 CART cells against multiple myeloma. This is good executed study. I have the following suggestion. Figure 6. Please add data on mouse weight in the study.

We appreciate the reviewer's positive evaluation of our study. Monitoring mouse weight is important in preclinical studies to assess any potential treatment-related toxicity or adverse effects. We have included this information to provide a more comprehensive view of the study's outcomes. From the results, we observed that in the MM1.S-xenograft mouse model, both the PBS and MOCK-CAR-T groups experienced a significant decline in mice body weight in the later stages of the experiment due to tumor growth. In contrast, the Bi-CAR-T group showed a steady increase in mice body weight. In the OPM2-xenograft mouse model, the body weight of mice in the MOCK-CAR-T group did not increase, while the Bi-CAR-T group exhibited a steady increase in body weight. These results suggest that Bi-CAR-T cells, while inhibiting MM growth, do not induce potential treatment-related toxicity.

Please refer to the following:

In revised Supplementary Fig. 17,

After Revision:

Supplementary Fig. 17. Body weight changes in MM-xenograft mouse models. (a) Mouse weight normalized to day 0 for each group, from the day after MM1.S cells injection (n = 5 per group). **(b)** Mouse weight normalized to day 0 for each group, from the day after OPM2 cells injection (n = 5 per group). One-way ANOVA was used for statistical analysis. $**P < .01$, ns = $P > .05$.

In "RESULTS" section, "Bispecific BCMA-CD24-CAR-T cells prolong MM mouse survival" part, page 15, lines 357 - 364.

After Revision:

Furthermore, in the MM1.S-xenograft mouse model, both the PBS and MOCK-CAR-T groups of mice experienced a significant decline in body weight in the later stages of the experiment due to tumor growth. In contrast, the Bi-CAR-T group showed a steady increase in mouse body weight. In the OPM2-xenograft mouse model, the body weight of mice in the MOCK-CAR-T group did not increase, while the Bi-CAR-T group exhibited a steady increase in body weight. These results suggest that Bi-CAR-T cells, while inhibiting MM growth, do not induce significant treatment-related toxicity (Supplementary Fig. 17).

Reviewer #2

The present study by Sun, Cheng et al. delves into the role of the CD24-Siglec10 immunological synapse between multiple myeloma cells and macrophages, and its potential as a therapeutic target for patients. The researchers focused on CD24 as a target and demonstrated that after anti-BCMA CAR-T treatment, the progression of the disease might be linked to slowly proliferating CD24-positive myeloma cells. To explore this further, they developed innovative anti-CD24 and bispecific anti-CD24/anti-BCMA CAR-T cells, and the results from in-vitro and xenograft mouse models are highly promising. Moreover, the study highlights an interesting finding related to macrophages. They observed an increased phagocytosis of myeloma cells by macrophages and changes in macrophage subpopulations, indicating a shift from M2 to M1 macrophages and a reversal of immune checkpoint signaling mediated by the blocking of CD24 on the myeloma cells.

Overall, the manuscript is well-written, follows a clear and logical structure, and represents the first study exploring the use of CD24+ CAR-T cells as a treatment strategy in multiple myeloma. The presented results are encouraging and hold great potential for further evaluation in clinical trials.

We appreciate the reviewer for the positive and comprehensive evaluation of our manuscript and the encouraging comments. The acknowledgment that our study “The presented results are encouraging and hold great potential for further evaluation in clinical trials.” means a lot to us.

To enhance the manuscript further, some revisions are suggested:

1. In lines 107ff, it's common practice to perform FDR correction of p-values for DEG analyses of scRNA sequencing data. Consider revising Fig 1E and the respective methods part (lines 390f.) to reflect this.

The reviewer raises a valuable suggestion. Performing False Discovery Rate (FDR) correction for p-values in scRNA sequencing data analysis is an important practice to control for multiple testing. We have changed the FDR-corrected p-values to Figure 1e and updated the methods section.

Please refer to the following:

In revised Fig. 1e,

After Revision:

Figure 1: Proportion of CD24-positive cells is increased in MM after treatment. (e) Volcano plot of the top differentially expressed genes (DEGs) in residual MM cells.

In “METHODS” section, “Patients Single-cell RNA sequencing (scRNA-seq) data process” part, page 20, lines 486 - 487.

After Revision:

Differentially expressed genes (DEGs) were denoted as statistically significant for the false discovery rate (FDR) less than 0.05 with a fold change exceeding 1.2.

2. Lines 124ff. Regarding the increased percentage of CD24+ MM cells after 24h incubation with bortezomib, it may not be appropriate to draw conclusions about drug resistance in such a short timeframe. Additional investigations into CD24 expression upregulation and testing with other drugs (e.g., dexamethasone, lenalidomide) could provide more insights.

The reviewer raised a valid and important point. Indeed, drug resistance in multiple myeloma is a complex process influenced by various factors, and describing this phenomenon as drug resistance after only a 24-hour co-incubation period may not be appropriate. We have corrected this description in the revised version.

It is an excellent suggestion to perform additional investigations to examine CD24 expression change and its relationship with dexamethasone and lenalidomide. We treated ten MM patients' BMMCs with 10 μ M dexamethasone (DEX) or 10 μ M lenalidomide (LEN) for 24 hours. The percentage of CD138⁺CD24⁺ cells increased in 8 of 10 BMMC samples post-DEX ($P = 0.03$). However, the percentage of CD138⁺CD24⁺ cells didn't change significantly in BMMC samples post-LEN ($P = 0.85$).

The reasons for this phenomenon may be related to NF- κ B signaling. Bortezomib is a proteasome inhibitor and its anti-MM action is partly mediated through inhibition of NF- κ B.^{1, 2, 3} Additionally, dexamethasone-induced apoptosis in MM cells is mediated by its initial binding to the glucocorticoid receptor, which is a ligand-activated transcription factor capable of downregulating NF- κ B upon activation.^{4, 5} Some studies have shown that CD24 expression efficiently attenuates NF- κ B signaling.^{6, 7} This suggested that cells with high CD24 expression may have weak NF- κ B signaling, making them less sensitive to NF- κ B inhibitors, resulting in their survival even after NF- κ B inhibitors treatment. On the other hand, lenalidomide is an immunomodulatory drug with a mechanism of action distinct from dexamethasone and bortezomib,^{8, 9} and it does not significantly affect the proportion of CD138⁺CD24⁺ cells. We plan to validate these hypotheses in future experiments.

Please refer to the following:

In revised Supplementary Fig. 3,

After Revision:

Supplementary Fig. 3. The frequency of CD138⁺CD24⁺ cells after dexamethasone or lenalidomide treatment. (a) Flow cytometry analysis of patient samples and the association between drug response and CD24 expression in MM cells (n =10). The frequency of the subpopulation of CD138⁺CD24⁺ cells increase in 8 of 10 primary myeloma samples post-dexamethasone (DEX) treatment. (b) The frequency of CD138⁺CD24⁺ cells didn't change significantly in BMCC samples post-lenalidomide (LEN) (n =10). Paired t-test was used. * $P < .05$, ns = $P > .05$.

In "RESULTS" section, "CD24-positive MM cells increase after BCMA-CAR-T treatment" part, page 7, lines 142 - 147.

After Revision:

Meanwhile, the other ten MM patients' BMMCs were treated with 10 μ M dexamethasone (DEX) or 10 μ M lenalidomide (LEN) for 24 hours. The frequency of CD138⁺CD24⁺ cells increased in 8 of 10 BMMC samples post-DEX ($P = 0.03$) (Supplementary Fig. 3a). However, the frequency of CD138⁺CD24⁺ cells did not change significantly in BMMC samples post-LEN ($P = 0.85$) (Supplementary Fig. 3b). These results indicate that CD24⁺ MM cells appear not to be sensitive to bortezomib and dexamethasone.

3. Lines 245f. Please revise wording here: The authors have shown that CD24-CAR-T activate phagocytosis and lead to differences in the macrophage subpopulations. However, for a full demonstration of changes in the signaling cascade, a confirmation on protein level would be preferable, e.g., Western Blotting showing phosphorylation status of SHP-1/2/ITIM regions or by proteomics. Moreover, the reference to Figure 4e seems to be misplaced. Fig 4e only shows that macrophages are present in different proportions in the BM and is neither related to function nor CD24/Siglec10 pathway.

Thanks for the valuable suggestion from the reviewer. The reviewer is correct. Figure 4e does not directly demonstrate changes in the CD24/Siglec10 pathway or macrophage function. We have corrected this in the revised version. Additionally, the reviewer's suggestion regarding the need for protein-level confirmation of changes in the signaling cascade is well-founded. To provide more robust evidence of these changes, we did the Western Blotting to assess the phosphorylation status of SHP-1/2.

We collected proteins from macrophages in the 5TGM1 mouse models and conducted Western blotting analysis. We found that the levels of phosphorylated SHP-1 and SHP-2 proteins were significantly elevated in the PBS group and MOCK group compared to the normal group.

However, in the CD24-CAR-T treatment group, the levels of phosphorylated SHP-1 and SHP-2 proteins were significantly reduced compared to the PBS group and MOCK group, although

they had not yet returned to the levels of the normal group. This indicates that CD24-CAR-T cells have a significant inhibitory effect on the phosphorylation of SHP-1 and SHP-2 proteins. These experiments would enhance the mechanistic understanding of how CD24-CAR-T cells modulate macrophage function.

Please refer to the following:

In revised Supplementary Fig. 12,

After Revision:

Supplementary Fig. 12. The expression levels of SHP-1/2-related proteins in macrophages. (a) Western Blotting analysis of the protein expression levels of Phospho-SHP-1 (Tyr564), total SHP-1 and total β -actin (n = 3 independent experiments). (b) Relative expression of Phospho-SHP-1 (Tyr564) or total SHP-1 to β -actin. (c) Western Blotting analysis of the protein expression levels of Phospho-SHP-2 (Tyr580), total SHP-2 and total β -actin (n = 3

independent experiments). (d) Relative expression of Phospho-SHP-2 (Tyr580) or total SHP-2 to β -actin. One-way ANOVA was used for statistical analysis. $**P < .01$, $***P < .001$, ns = $P > .05$.

In “RESULTS” section, “CD24-CAR-T cells modulate myeloma microenvironment” part, page 12, lines 276.

After Revision:

CD24-CAR-T cells clearly affect macrophages (Fig. 4c).

In “RESULTS” section, “CD24-CAR-T cells modulate myeloma microenvironment” part, page 12 - 13, lines 290 - 299.

After Revision:

To further validate the impact of CD24-CAR-T cells on the CD24/Siglec-10 signaling at the protein level, we collected proteins from macrophages in the 5TGM1 mouse models and conducted Western blot analysis to assess the phosphorylated protein levels of SHP-1 and SHP-2. In comparison to the normal group, the levels of phosphorylated SHP-1 and SHP-2 proteins were significantly elevated in both the PBS group and the MOCK group. However, in the CD24-CAR-T treatment group, the levels of phosphorylated SHP-1 and SHP-2 proteins were notably reduced compared to the PBS group and MOCK group, although they had not completely returned to the levels observed in the normal group (Supplementary Fig. 12). This suggests that CD24-CAR-T cells exert a significant inhibitory effect on the phosphorylation of SHP-1 and SHP-2 proteins within the CD24/Siglec-10 signaling pathway.

In “METHODS” section, “Western Blotting for phosphorylation status of SHP-1/2” part, page 25, lines 617 - 632.

After Revision:

Western Blotting for phosphorylation status of SHP-1/2

Mouse macrophages were isolated from BMMCs of 5TGM1/KaLwRij MM mouse models with CD11b MicroBeads (Miltenyi). Mouse macrophages were lysed in the mammalian cell extraction buffer (BioVision) with Protease and Phosphatase Inhibitors (ThermoFisher). Protein lysates were incubated on ice for 10 min and centrifuged at 13,000 rpm for 10 min at 4 °C. Proteins were separated with 4% - 12% Bis-Tris Gel (Invitrogen) at 120 V for 1.5 h. And then transferred to a nitrocellulose (NC) membrane for 1.5 h at 200 mA. The membrane was blocked for 1.5 h with 5% milk at room temperature. Primary antibodies SHP-1 (Cell Signaling Technology, catalog # 3759, clone # C14H6, 1:1000), SHP-2 (Cell Signaling Technology, catalog # 3397, clone # D50F2, 1:1000), Phospho-SHP-1 (Tyr564) (Cell Signaling Technology, catalog # 8849, clone # D11G5, 1:1000), Phospho-SHP-2 (Tyr580) (Cell Signaling Technology, catalog # 5431, clone # D66F10, 1:1000), β -Actin (Cell Signaling Technology, catalog # 4967, 1:1000), were incubated overnight at 4 °C. Anti-rabbit IgG, HRP-linked antibody (Cell Signaling Technology, catalog # 7074, 1:1000) were incubated for 2 h. ECL Substrates (Bio-Rad, catalog # 1705060) were used for exposure. Western Blotting imaging was exposed with a Bio-Rad ChemiDoc XRS⁺. The images were analyzed by Image Lab Software (Bio-Rad).

4. Lines 248ff. and Fig 4f. What are the percentual changes between the macrophage clusters? Do the percentages differ? The authors state “Cluster 0 was increased in MM-bearing mice, but decreased to normal levels after CD24-CAR-T cells treatment.” However, the total amount of cells captured after CD24-CAR-T is less compared to PBS and MOCK-CAR-T conditions making it difficult to read if the differences between conditions are also reproducible in relative numbers.

Thanks for the valuable feedback from the reviewer. Since the baseline capture cell numbers vary among the normal group, PBS group, MOCK-CAR-T group, and CD24-CAR-T group, it

may not be appropriate to compare them directly using relative numbers. Instead, we should use percentual changes to compare intergroup differences.

In the revised version, we reevaluated the data and provided the specific percentage changes between macrophage subclusters. From the results of the reanalysis, the previous description of Cluster 0 was inaccurate. In comparison to the normal group, there is a significant increase in the proportion of cells in Cluster 0 in both the PBS group and the MOCK group. However, in the CD24-CAR-T treatment group, the proportion of cells in Cluster 0 is significantly reduced compared to the PBS group and MOCK group, although it has not yet returned to the levels seen in the normal group. Therefore, in the revised version, we have removed the 'but decreased to normal levels after CD24-CAR-T cells treatment' and provided a new description. We have also included the cell proportion graph of subclusters of macrophages in Figure 4. This suggestion helps ensure that the differences between conditions are accurately represented in terms of relative numbers.

Please refer to the following:

In revised Fig. 4e,

After Revision:

Figure 4: scRNA-seq revealed tumor microenvironment modification after CD24-CAR-T cells treatment. (e) Bar-views showed the proportion of cells in subclusters of macrophages of

healthy and MM-bearing mice treated with PBS, MOCK-CAR-T, or CD24-CAR-T cells. * $P < .05$, ** $P < .01$, *** $P < .001$, ns = $P > .05$.

In “RESULTS” section, “CD24-CAR-T cells modulate myeloma microenvironment” part, page 12, lines 276 - 283.

After Revision:

We analyzed the proportion of cells in subclusters of macrophages and discovered that two subclusters were dramatically altered in mice treated with CD24-CAR-T cells compared to the controls (Fig. 4d, e). In comparison to the normal group, there is a significant increase in the proportion of cells in Cluster 0 in both the PBS group and the MOCK group. However, in the CD24-CAR-T treatment group, the proportion of cells in Cluster 0 is significantly reduced compared to the PBS group and MOCK group, although it has not yet returned to the levels seen in the normal group. Cluster 4 was a new cluster that appeared after CD24-CAR-T cells treatment.

5. In lines 251f, the statement "The differentially expressed genes were divided into upregulated genes and downregulated genes" is trivial and can be deleted.

Thanks for the reviewer's suggestion to improve the clarity and conciseness of the manuscript. We removed the statement "The differentially expressed genes were divided into upregulated genes and downregulated genes" from lines 251f in the revised manuscript. This change helps streamline the text and eliminate redundancy.

Please refer to the following:

In “RESULTS” section, “CD24-CAR-T cells modulate myeloma microenvironment” part, page 12, lines 283 - 285.

After Revision:

We identified differentially expressed genes between these two clusters and compared those with the other clusters. Pathway analysis was performed for these genes separately.

6. In lines 276ff, emphasis should be placed on comparing lysis rates between BI-CAR-T and single CD24-CAR-T cells, rather than comparing with MOCK-CAR-T, which is trivial and bears no new insights.

Thanks for the reviewer's valuable suggestion. We have revised the comparison of lysis rates between Bi-CAR-T and CD24-CAR-T cells. The reviewer is correct. If we only focus on the comparison between Bi-CAR-T and MOCK-CAR-T, the results bear no new insights. We should place more emphasis on the advantages of Bi-CAR-T compared to CD24-CAR-T. This change helps focus the discussion on the relevant comparisons and provides a clearer interpretation of the results.

Please refer to the following:

In "RESULTS" section, "Bispecific CAR-T cells exhibit increased cytolytic activity in vitro" part, page 14, lines 330 - 336.

After Revision:

When the ratio of Bi-CAR-T cell : MM cell was 5:1, 87.3% of target cells were lysed in the MM1.S cell line, and there was a 1.3-fold increase in lysis of MM cells with Bi-CAR-T cells compared to CD24-CAR-T cells. In the OPM2 cell line, 94.2% of target cells were lysed, and there was a 1.3-fold increase in lysis of MM cells with Bi-CAR-T cells compared to CD24-CAR-T cells. In the H929 cell line, 93.0% MM cells were lysed by Bi-CAR-T cells, and there was a 2.1-fold increase compared to the CD24-CAR-T cells (Fig. 5c).

7. Regarding the lysis capacity differences between cell lines (lines 276ff / Fig. 5), consider discussing the BCMA expression levels of these cell lines, as variations in BCMA expression may contribute to the differences observed.

The reviewer raised a valuable and important point. Indeed, the abundance of antigens on the surface of target cells plays a crucial role in the lysis of target cells by CAR-T cells. Considering the variations in BCMA expression levels among the MM1.S, OPM2, and H929 cell lines is a valid point that can contribute to the observed differences in lysis capacity. We have previously assessed the expression levels of BCMA in different MM cell lines. We found that BCMA expression levels are generally high in MM cell lines. In the MM1.S, OPM2, and H929 cell lines used in this study, the BCMA expression levels are all above 90%. Please refer to the flow cytometry plot and bar graph below for details.

Since the expression levels of BCMA are high in the MM1.S, OPM2, and H929 cell lines, we observed that there was no significant difference in the lysis capacity between the BCMA-CAR-T and Bi-CAR-T groups when targeting these three cell lines. Please refer to the bar graph below for details.

We have included the information of BCMA expression levels in the Results section. This additional information will provide a more comprehensive interpretation of the results.

Please refer to the following:

In "RESULTS" section, "Bispecific CAR-T cells exhibit increased cytolytic activity in vitro" part, page 14, lines 327 - 330.

After Revision:

Due to the high expression levels of BCMA in MM1.S (96.6%), OPM2 (91.3%), and H929 (99.5%) cell lines (Supplementary Fig. 15a), CAR-T cells targeting BCMA have demonstrated high MM cell lysis capabilities. The cytolytic activity results showed that compared with other groups, the Bi-CAR-T cells had the highest cytolytic activity in vitro.

In revised Supplementary Fig. 15a,

After Revision:

Supplementary Fig. 15. BCMA expression in MM cell lines and bispecific CAR-T cells

release TNF- α in vitro. (a) Bar plot showing expression levels of BCMA in MM1.S, OPM2, and H929 cell lines.

8. In line 325, correct the typo: change "ant-CD24" to "anti-CD24."

The reviewer's attention to detail is appreciated. We apologize for the mistake, which has now been corrected.

Please refer to the following:

In "DISCUSSION" section, page 16, lines 386 - 387.

After Revision:

SWA11 is a high-affinity anti-CD24 antibody with strong cytotoxic activity and low off-target toxicity.

9. Lines 324ff. Are there pre-clinical or clinical studies of SWA11 in the setting of multiple myeloma? If not, I would add this information to elevate the discussion.

Thanks for the reviewer's valuable suggestion. We apologize for not providing detailed information about SWA11. The performance of the SWA11 antibody, which is an essential component of the CD24-CAR, in preclinical or clinical settings is crucial information. So far, no research team has used SWA11 in clinical trials for multiple myeloma. However, some research teams have demonstrated promising results in preclinical studies for multiple myeloma using SWA11.¹⁰ In the revised version, we added information about SWA11 to elevate the discussion.

Please refer to the following:

In "DISCUSSION" section, page 16, lines 387 - 393.

After Revision:

To date, there is no report of SWA11 being used in clinical trials for multiple myeloma. However, research teams have demonstrated the favorable effects of SWA11 in preclinical studies. Specifically, the combination of SWA11 with bortezomib (BTZ) resulted in an extension of the survival of multiple myeloma mice.¹⁰ Additionally, Schambach *et al* developed SWA11-based anti-CD24 CAR-NK cells with cytotoxicity against ovarian cancer cell lines and patient-derived ovarian cancer cells.¹¹

10. In lines 353ff, the statement "CD24 is rarely expressed in normal human tissues" contradicts data from the Human Protein Atlas. Here, CD24 has high protein expression scores among various tissues including the CNS, lungs, intestines and reproductive organs. The data regarding potential toxicity presented in the study is just restricted to liver, kidney and spleen. We would highly recommend to revise this statement and – if no further data is available to show otherwise – to include a limitations section discussing adverse events, that might be encountered. The high expression in brain is particularly worrisome, specifically in the context of CAR-T-mediated neurotoxicity.

We thank the reviewer for bringing this to our attention. The reviewer is correct. Based on data from the Human Protein Atlas our statement in the manuscript, "CD24 is rarely expressed in normal human tissues," is inaccurate. We have modified this statement in the revised version. Additionally, in the revised version, we discussed the potential adverse events, particularly CAR-T-mediated neurotoxicity, and proposed potential mitigation strategies. This would provide a more balanced perspective on the safety profile of CD24-CAR-T cells.

Please refer to the following:

In "DISCUSSION" section, page 18 - 19, lines 443 - 456.

After Revision:

In CAR-T cell treatment, safety is one of the major challenges, including precise tumor targeting to avoid off-target or on-target/off-tumor toxicity.^{12, 13} CD24 is expressed on various hematopoietic cells, such as B cells¹⁴ and eosinophils¹⁵, and is also found in non-hematopoietic cells, including neural cells¹⁶, epithelial cells¹⁷, pancreatic cells¹⁸, and some cancer cell types.¹⁹ This inevitably raises concerns about the safety of CD24-CAR-T therapy. After all, neurotoxicity has always been an issue that needs to be addressed and prevented in CAR-T treatments.²⁰ To promptly terminate the function of CAR-T in case of adverse reactions, we integrated RQR8, a suicide molecule, as an immunological safety switch into our design.²¹ We have not found safety problems in our *in vivo* studies and we are conducting further research to avoid off-target toxicity. Meanwhile, to eliminate off-target effects at the source, we are constructing a CAR-T that will permit activation of CD24-targeted CARTs only in a MM microenvironment. This inducible CAR-T design relies on an autocleavable receptor construct (synNotch)²² that activates CD24-targeted CAR expression upon recognition of MM specific makers (e.g. CD38, SLAMF7) on the surface of myeloma cells.

Reviewer #4

General comment:

First I am delighted to serve as a reviewer for this work. In this work, Sun and Cheng et al developed CD24-CAR-T cells and analyzed in-vitro and in vivo data to assess the killing ability of MM cells. They further investigated the tumor microenvironment changes, particularly macrophage phenotypic alterations following CAR-T treatment. Finally, the authors constructed bispecific CAR-T cells, BCMA-CD24-CAR-T. The subsequent functional analysis, which showcases enhanced efficacy compared to monospecific BCMA-CAR-T-cell therapy, holds immense clinical significance.

Overall the study demonstrated a large amount of work; nevertheless, there are several areas that require significant improvement. Primarily, the data analysis of the single-cell RNA sequencing (scRNAseq) segment appears to be incomplete and lacks the necessary depth to provide comprehensive insights. Additionally, the organization of the manuscript needs improvement, as it currently lacks a cohesive and well-connected structure.

We thank the reviewer for the favorable assessment of our manuscript. The acknowledgment means a lot to us.

My major concerns are stated as the following:

1. I observed CD3D expression in part of Plasma/MM clusters in the umap of supplementary figure 1, any chances that these are B and T cell doublets?

We appreciate the reviewer's thorough observation, which made us aware of the issue with doublets. Doublets can occur in single-cell RNA sequencing data when two or more cells are captured together in a single droplet during library preparation. As a result, the expression profile of a doublet cell can contain genes from multiple cell types. This is a valid point, prompting us to re-examine the scRNA-seq code we have used. Upon reevaluating the code, we found that we utilized Seurat's integration workflows for quality control. One of the steps involved the removal of predicted doublets. Therefore, we thought that doublets should not be present in our samples.

We were also curious about the identity of these cells, so we searched CD3D using the Human Protein Atlas database (<https://www.proteinatlas.org/>). We discovered that, indeed, besides T cells, there are some B cells and Plasma cells expressing CD3D (please see Figure a below). Additionally, single cell bone marrow overview through the Human Protein Atlas also confirmed the expression of CD3D in some B cells and Plasma cells (please see Figure b below).

Figure legend: Analysis of CD3D expression in different cell types using the Human Protein Atlas database. (a) RNA expression of CD3D in different cell types.

(<https://www.proteinatlas.org/ENSG00000167286-CD3D/single+cell+type>) **(b) Single-cell**

sequencing analysis of CD3D expression in different cell types in the bone marrow. (<https://www.proteinatlas.org/ENSG00000167286-CD3D/single+cell+type/bone+marrow>)

The results displayed in the Human Protein Atlas database were consistent with our analysis, so we believe that this particular group of cells is indeed not doublets. In the revised version, we have refined the quality control methods, making our article even more rigorous.

Please refer to the following:

In “METHODS” section, “Patients Single-cell RNA sequencing (scRNA-seq) data process” part, page 19, lines 471 - 476.

After Revision:

The raw scRNA-seq data were loaded into R through the Seurat V4 package. The Seurat's integration workflows were used for quality control. The cells that have fewer than 200 genes, greater than 7,000 genes, and more than 10% of unique molecular identifiers stemming from mitochondrial genes were discarded from the analysis. Predicted doublets were removed. For individual samples, principal component analysis (PCA) was then performed on significantly variable genes of the remaining high-quality cells.

2. Related to figure 1e, a boxplot showing the expression level of CD24 of all patients' MM cells in D28 vs. Pre-infusion would be very nice to help the readers to get an overall expression difference.

We appreciate the valuable suggestion provided by the reviewer. Box plots or violin plots, in comparison to volcano plots, can offer readers a more intuitive understanding of the overall expression differences of a particular gene and whether there is a statistically significant difference in expression.

We conducted an analysis of the expression level of CD24 in all patients' MM cells at D28 vs. Pre-infusion and created both a box plot and a violin plot. We found that in the Pre-infusion group, there were relatively few cells expressing CD24, constituting only 3.8% (498 out of 13,108 cells). In contrast, at D28, the number of cells expressing CD24 significantly increased, reaching 24.1% (111 out of 460 cells) (please see Figure below).

Figure legend: (a) The box plot of CD24 expression in each pre-infusion MM and post-infusion D28 MM cells. (b) The violin plot of CD24 expression in each pre-infusion MM and post-infusion D28 MM cells. *** $P < .001$.

As most cells did not express CD24, the main body of the boxplot is centered around 0. Violin plot, being more intuitive compared to box plot, was employed in the revised version to illustrate the differences in CD24 expression between the D28 and Pre-infusion groups.

Please refer to the following:

In “RESULTS” section, “CD24-positive MM cells increase after BCMA-CAR-T treatment” part, page 6, lines 124 - 126.

After Revision:

Based on the violin plot, it is evident that the proportion of CD24-positive MM cells in the post-infusion D28 group significantly increases when compared to the pre-infusion group. (3.8% vs 24.1%, $P < 0.001$) (Fig. 1f).

In revised Fig. 1f,

After Revision:

Figure 1: Proportion of CD24-positive cells is increased in MM after treatment. (f) The violin plot of CD24 expression in each pre-infusion MM and post-infusion D28 MM cells. * $P < .001$.**

3. It would be more robust if the authors could include additional public scRNA-seq cohorts showing the either increased expression of CD24 or increased expression frequency of CD24 in MM cells.

The reviewer raises a valuable suggestion. Incorporating additional public scRNA-seq cohorts for validation would enhance the robustness of our findings. This approach provides

independent validation and strengthens our assertions concerning the significance of CD24 in multiple myeloma.

We reviewed the manuscripts related to scRNA-seq of pre-treatment and post-treatment samples in BCMA-CAR-T therapy. We found a study entitled "Identification of potential resistance mechanisms and therapeutic targets for the relapse of BCMA CAR-T therapy in relapsed/refractory multiple myeloma through single-cell sequencing."²³ This study used 10X Genomic scRNA-seq to identify cell populations in relapsed/refractory multiple myeloma (RRMM) CD45⁺ BM cells before BCMA CAR-T treatment and at relapse after BCMA CAR-T treatment. However, the samples used in this study (CD45⁺ BM cells enriched with human CD45 microbeads) differ from the samples analyzed in our study (bone marrow mononuclear cells isolated using density gradient centrifugation). Therefore, we did not integrate those for combined analysis. However, the results from this study can serve as a reference. This paper indicated that CD24 was found to be expressed in myeloma cells at the point of relapse following BCMA CAR-T cell therapy and the percentage of plasma cells with BCMA, CD38, CD24, CD138, SLAMF7, and GPRC5D expression increased at relapse after BCMA CAR-T cell therapy compared to baseline.

This external data provides support for our conclusions. In the revised version of our manuscript, we have emphasized the findings from this study.

Please refer to the following:

In "INTRODUCTION" section, page 4 -5, lines 88 - 91.

After Revision:

Li W, *et al* employed single-cell RNA sequencing (scRNA-seq) to delineate cell populations within CD45⁺ bone marrow cells of patients with RRMM both before and after BCMA CAR-T treatment. Their findings indicated that CD24 was found to be expressed in myeloma cells at relapse following BCMA CAR-T cell therapy.²³

In "DISCUSSION" section, page 16, lines 383 - 384.

After Revision:

Furthermore, it was observed that more myeloma cells expressed more frequently CD24 upon relapse subsequent to BCMA CAR-T cell therapy.²³

4. Related with figure 1h, is CD24 expression level also increased in MM after Bortezomib treatment? or the treatment only affected the frequency of CD24+ MMs?

The reviewer raised an important point. The increase in the frequency of CD24+ cells and the increase in the cell surface CD24 expression levels are two distinct outcomes, each leading to different conclusions. Our flow cytometry results indicated an increase in the frequency of CD24+ cells. The mean fluorescence intensity (MFI) of cells did not increase, demonstrating that cell surface CD24 expression levels did not change a lot (please see Figure below).

Figure legend: Flow cytometry analysis of patient samples and the association between drug response and CD24 expression in MM cells. The mean fluorescence intensity (MFI) of the subpopulation of CD138⁺CD24⁺ cells. Paired t-test was used. ns = $p > .05$.

The reasons for this phenomenon may be related to NF- κ B signaling. Bortezomib is a proteasome inhibitor. Its anti-MM action is partly mediated through inhibition of NF- κ B.^{1, 2, 3} Some studies have shown that CD24 expression efficiently attenuates NF- κ B signaling.^{6, 7} This suggested that cells with CD24 expression may have weak NF- κ B signaling, making them less sensitive to NF- κ B inhibitors, resulting in their survival even after NF- κ B inhibitors treatment. We plan to validate these hypotheses in future experiments.

In the revised version, we have clarified that Figure 1i (originally Figure 1h) represents an increase in the frequency of CD24⁺ cells.

Please refer to the following:

In “RESULTS” section, “CD24-positive MM cells increase after BCMA-CAR-T treatment” part, page 6 - 7, lines 140 - 141.

After Revision:

We treated fifteen MM patients' BMBCs with 5 nM bortezomib (BTZ) for 24 hours. The frequency of CD138⁺CD24⁺ cells increased in 12 of 15 BMBC samples post-BTZ (Fig. 1i).

In revised Fig. 1i,

After Revision:

Figure 1: Proportion of CD24-positive cells is increased in MM after treatment. (i) Flow cytometry analysis of patient samples and the association between drug response and CD24 expression in MM cells (n = 15). The frequency of the subpopulation of CD138⁺CD24⁺ cells increase in 12 of 15 primary myeloma samples post-bortezomib (BTZ) treatment. Paired t-test was used. **P* < .05

5. In figure 2h and 2i, MOCK-CAR-T show similar phagocytosis level to PBS. Did the authors compared the phagocytosis level by using other MM cell lines with different CD24 expression %? For example, MM1.S or H929?

The reviewer raised a valid point. To assess the generalizability of the findings regarding phagocytosis and CD24 expression, it is important to include experiments with multiple MM cell lines that vary in their CD24 expression levels. This helps confirm whether the observed effects on phagocytosis are specific to the cell line used in the study or if they can be extended to other MM cell lines with different CD24 expression profiles.

In the revised version, we added experiments involving the phagocytosis of MM1.S-GFP cells and H929-GFP cells by macrophages. MM1.S-GFP cells or H929-GFP cells, DiD-stained macrophages, and CAR-T cells were co-cultured at a ratio of 2:1:1 for 4 hours in the 37 °C incubator. Next, cells were washed with PBS twice, and GFP+DiD+ cells were detected by BD FACSVerser™ flow cytometry and ZEISS Axio Observer fluorescence microscope.

Comparing the phagocytosis rates of the three cell types in the PBS group and MOCK-CAR-T group, it's evident that the proportion of MM1.S cells (PBS: 3.02%; MOCK: 3.09%) and H929 cells (PBS: 3.32%; MOCK: 3.52%) that were phagocytized increased compared to OPM2 cells (PBS: 1.44%; MOCK:1.75%). This might be related to the differential surface expression levels of CD24 in these three cell lines (OPM2: 28.3%; MM1.S: 11.7%; H929: 7.77%). This observation aligns with the concept of CD24 acting as a "Don't Eat Me" signal. Importantly, a consistent observation across all three cell lines was that when compared with the MOCK-CAR-T cells and BCMA-CAR-T cells, the Bi-CAR-T cells significantly enhanced phagocytic clearance by macrophages, similar to the CD24-CAR-T cells.

Please refer to the following:

In revised Supplementary Fig. 16,

After Revision:

Supplementary Fig. 16. CD24-CAR-T and Bi-CAR-T cells promote phagocytic clearance

by macrophages in vitro. (a) Phagocytosis was performed by coculture of MM1.S cells that expressed GFP (green), DiI-stained macrophages (red), and CAR-T cells at a ratio of 2:1:1. After a 4-hour coculture, suspended cells were washed and detected. Fluorescent images of phagocytic clearance (n = 3 independent experiments). **(b)** Bar plot showing the percentage of

MM1.S phagocytosis detected by flow cytometry analysis (n = 3 independent experiments). (c)
Phagocytosis was performed by coculture of H929 cells (n = 3 independent experiments). (d)
Bar plot showing the percentage of H929 phagocytosis detected by flow cytometry analysis
(n = 3 independent experiments). One-way ANOVA was used for statistical analysis * $P < .05$,
** $P < .01$, *** $P < .001$, ns = $P > .05$.

In “RESULTS” section, “Bispecific CAR-T cells exhibit increased cytolytic activity in vitro” part,
page 14, lines 341 - 344.

After Revision:

Compared with the MOCK-CAR-T cells and BCMA-CAR-T cells, the Bi-CAR-T cells strongly
promoted phagocytic clearance by macrophages, similar to the CD24-CAR-T cells (Fig. 5h and
Supplementary Fig. 16a, c). Flow cytometry also showed the Bi-CAR-T cells significantly
increased the phagocytic clearance by macrophages (Fig. 5i and Supplementary Fig. 16b, d).

In “METHODS” section, “Macrophage phagocytosis” part, page 23, lines 562 - 564.

After Revision:

In vitro phagocytosis was performed by coculture of OPM2, MM1.S, or H929 cells that
expressed GFP, DiD stained macrophages, and CAR-T cells at a ratio of 2:1:1 for 4 hours in the
37 °C incubator.

6. Regarding the single cell RNAseq analysis in figure4, did the author observed other celltypes
except the lymphoid/myeloid cell populations shown in the figure?

The reviewer raised an important point. In this experiment, the cell populations in the mouse
bone marrow should include not only lymphoid and myeloid cells but also erythroid cells, the

injected myeloma cells (5TGM1), the injected CAR-T cells, and a small number of hematopoietic stem cells (HSCs).

The purpose of single cell RNAseq analysis was to investigate changes in the bone marrow microenvironment. To achieve this, we removed the artificially injected 5TGM1 myeloma cells and CAR-T cells. Moreover, since the 5TGM1 myeloma cells and CAR-T cells constituted a substantial proportion of the sample (in the PBS group and MOCK group, MM cells accounted for approximately 35% of the total), this would have interfered with the analysis of the bone marrow microenvironment cells. We processed mouse bone marrow cells as follows. After extracting bone marrow from the mice, we performed Ficoll-Paque density gradient centrifugation to isolate bone marrow mononuclear cells (BMMCs). This step involved the removal of a significant number of mature red blood cells (without nucleus), leaving only a small population of erythroblasts. We employed fluorescence-activated cell sorting (FACS) to separate the 5TGM1 and CAR-T cells, retaining only the cells associated with the bone marrow microenvironment.

As to Supplementary Fig. 10a, the final UMAP primarily consists of lymphoid and myeloid cells, along with a small proportion of erythroblasts (Cluster 16). HSCs may not have been detected due to their low frequency in the sample.

7. Did the author performed cell cell communication analysis using CAR-T cells and macrophages captured in scRNAseq? What are the most involved communication signals in treatment group?

Thanks for this valuable suggestion. Cell-cell communication analysis can be a valuable approach to understand the signaling pathways and interactions between different cell types within a complex microenvironment. This information would be important in elucidating the

mechanisms of CAR-T cell therapy and their impact in the bone marrow microenvironment. We have also performed cell-cell communication analysis in the revised version of the manuscript. However, as explained in our previous response for Question 6, we have excluded MM cells and CAR-T cells from our sample. Therefore, our analysis does not involve the communication between MM cells and CAR-T cells with other cell types. Based on our in vitro results, it is expected that MM cells have the most communication with CAR-T cells.

Through cell-cell communication analysis of the CD24-CAR-T treatment group, we found that the highest number of interactions occurred within monocytes and monoblasts , and interactions between monocytes and monoblasts. Regarding interaction weights or strength, the most prominent interactions were observed within neutrophils and between neutrophils and myelocytes. In the CD24-CAR-T treatment group, the most involved communication signals were related to the Galectin-9 (Lgals9) signaling pathway. The most critical sources and targets of these signals were a series of neutrophils and a series of monocytes. It has been reported that Galectin-9 is upregulated in murine M1 macrophages,²⁴ which aligns with our flow cytometry results indicating a significant increase in M1 macrophages after CD24-CAR-T treatment.

Please refer to the following:

In revised Supplementary Fig. 13,

After Revision:

Supplementary Fig. 13. Cell-cell communication analysis of CD24-CAR-T treatment group. (a) Circle plots of the number of interactions and interaction strength of CD24-CAR-T treatment group. (b) Heatmaps of the interactions and interaction strength of CD24-CAR-T treatment group. (c) Hierarchy plots showing the most involved communication signal in CD24-CAR-T treatment group.

In “RESULTS” section, “CD24-CAR-T cells modulate myeloma microenvironment” part, page 13, lines 304 - 313.

After Revision:

Through cell-cell communication analysis of the CD24-CAR-T treatment group using scRNAseq, we found that the highest number of interactions occurred within monocytes and monoblasts, and interactions between monocytes and monoblasts. Regarding interaction weights or strength, the most prominent interactions were observed within neutrophils and between neutrophils and myelocytes (Supplementary Fig. 13a, b). In the CD24-CAR-T treatment group, the most pronounced communication signals were related to the Galectin-9 (Lgals9) signaling pathway. The most critical sources and targets of these signals were neutrophils and monocytes (Supplementary Fig. 13c). It has been reported that Galectin-9 is upregulated in murine M1 macrophages,²⁴ which aligns with our flow cytometry results indicating a significant increase in M1 macrophages after CD24-CAR-T treatment.

8. How did the authors perform celltype annotation? I couldn't find the detailed description in the method part. Did the author applied independent cell type annotation tools, including celltypist, singleR etc to validate the robustness of celltype annotation for scRNAseq results shown in figure 1b and in figure 4a&b? Relying solely on a single annotation result may introduce vulnerabilities and potential biases in the analysis.

The reviewer raised an important point. Depending solely on a single annotation result, can introduce vulnerabilities and potential biases into the analysis. We used the following methods for cell annotation: First, the gene-specific markers for each cluster were determined using the FindMarkersAll function with MAST test statistics. Subsequently, the top 20 gene-specific markers were input into the CellMarker 2.0 cell annotation tool²⁵ to obtain automatic

annotations. Then, cluster annotations were generated through a combination of automatic annotation and manual annotation based on relevant studies ^{26, 27}.

We apologize for not providing detailed information about celltype annotation. We have included this information in the revised version.

Please refer to the following:

In “METHODS” section, “Patients Single-cell RNA sequencing (scRNA-seq) data process” part, page 20, lines 479 - 484.

After Revision:

The gene-specific markers for each cluster were determined using the FindMarkersAll function with Model-based Analysis of Single-cell Transcriptomics (MAST) test statistics. Subsequently, the top 20 gene-specific markers were input into the CellMarker 2.0 cell annotation tool ²⁵ to obtain automatic annotations. Then, cluster annotations were generated through a combination of automatic annotation and manual annotation based on relevant studies ^{26, 27}.

9. The authors should evaluate batch effects by employing appropriate computational methods like ComBat, Harmony, or Seurat's integration workflows.

Thanks for the reviewer’s valuable suggestion. Batch effects can arise when scRNA-seq data are generated in different batches or runs, and if not corrected, they can confound the analysis and interpretation of results. Addressing batch effects is essential in scRNA-seq studies to ensure that observed differences between groups or conditions are not due to technical variations introduced by different data acquisition batches.

In our study, the Seurat's integration workflows were used for quality control. Before integrating the data, we visualized the batch effects. We used principal component analysis (PCA) and plot the principal components, which can reveal batch-related clustering. From the PCA plot below, we can see cells from different samples are intermingled. There is no strong segregation of cells

from different samples. (Cluster 8 and 9 are primarily concentrated in normal samples, so they are not segregated cell populations.) So, the samples we used did not appear to exhibit batch effects.

Minor comments:

1. Minor comment: line 245-246: I didn't see evidence for blocking of cd24/siglec-10 signaling in figure 4e. suggest to move downwards after introducing results of figure 4f,g and h.

Thanks for the valuable suggestion from the reviewer. Figure 4e does not directly demonstrate changes in the CD24/Siglec10 pathway or macrophage function. We have moved this description downwards after introducing results of figure 4f-h in the revised version. Additionally, to provide more robust evidence of these changes, we performed Western Blotting to assess the phosphorylation status of SHP-1/2.

We collected proteins from macrophages in the 5TGM1 mouse models and conducted Western blotting analysis. We found that the levels of phosphorylated SHP-1 and SHP-2 proteins were significantly elevated in the PBS group and MOCK group compared to the normal group.

However, in the CD24-CAR-T treatment group, the levels of phosphorylated SHP-1 and SHP-2 proteins were significantly reduced compared to the PBS group and MOCK group, although they had not yet returned to the levels of the normal group. This indicated that CD24-CAR-T cells have a significant inhibitory effect on the phosphorylation of SHP-1 and SHP-2 proteins. These experiments would enhance the mechanistic understanding of how CD24-CAR-T cells modulate macrophage function.

Please refer to the following:

In revised Supplementary Fig. 12,

After Revision:

Supplementary Fig. 12. The expression levels of SHP-1/2-related proteins in macrophages. (a) Western Blotting analysis of the protein expression levels of Phospho-SHP-1 (Tyr564), total SHP-1 and total β -actin (n = 3 independent experiments). (b) Relative expression of Phospho-SHP-1 (Tyr564) or total SHP-1 to β -actin. (c) Western Blotting analysis of the protein expression levels of Phospho-SHP-2 (Tyr580), total SHP-2 and total β -actin (n = 3

independent experiments). (d) Relative expression of Phospho-SHP-2 (Tyr580) or total SHP-2 to β -actin. One-way ANOVA was used for statistical analysis. $**P < .01$, $***P < .001$, ns = $P > .05$.

In “RESULTS” section, “CD24-CAR-T cells modulate myeloma microenvironment” part, page 12, lines 276.

After Revision:

CD24-CAR-T cells clearly affect macrophages (Fig. 4c).

In “RESULTS” section, “CD24-CAR-T cells modulate myeloma microenvironment” part, page 12 - 13, lines 290 - 299.

After Revision:

To further validate the impact of CD24-CAR-T cells on the CD24/Siglec-10 signaling at the protein level, we collected proteins from macrophages in the 5TGM1 mouse models and conducted Western blot analysis to assess the phosphorylated protein levels of SHP-1 and SHP-2. In comparison to the normal group, the levels of phosphorylated SHP-1 and SHP-2 proteins were significantly elevated in both the PBS group and the MOCK group. However, in the CD24-CAR-T treatment group, the levels of phosphorylated SHP-1 and SHP-2 proteins were notably reduced compared to the PBS group and MOCK group, although they had not completely returned to the levels observed in the normal group (Supplementary Fig. 12). This suggests that CD24-CAR-T cells exert a significant inhibitory effect on the phosphorylation of SHP-1 and SHP-2 proteins within the CD24/Siglec-10 signaling pathway.

Reviewer #5

Sun and colleagues have submitted a manuscript, where they identified CD24 as a marker of relapsing multiple myeloma cells after anti-BCMA CAR therapy and hypothesized that CD24 may also play a role in immune evasion of MM in this context. They went on to design new CAR towards CD24 and dual CD24-BCMA CAR. They demonstrate activity in vitro and in vivo and hypothesise that CD24-CAR may be particularly acting through macrophages and show macrophage reprogramming in vivo. As such the concept is novel and innovative. The paper is well written and easy to follow. In the current status, I have concerns with several of the conclusions, which I do not think they are fully backed by data, as follows:

We thank the reviewer for the positive evaluation of our concept and the encouraging comments.

1. For their experiments, the authors claim to use cell lines with low expression of CD24 (range 7-28%). Still when performing killing experiment most cells are lysed, which raises concerns as to the specificity of the CAR. While the authors have used CD24 negative cells (HEK); in the light of this data MM CD24KO should be used.

The reviewer raises a valuable point. This is an important point to consider, as demonstrating the CAR's specificity for CD24-positive multiple myeloma cells is crucial for its clinical relevance. In vitro study we found that as the ratio 5 : 1 of CD24-CAR-T versus MM cells, lysed MM cells were 78.9%, 65.0% and 38.7% in the OPM2, MM1.S and H929 cell lines, respectively. The CD24⁺ proportions for the three cell lines were as follows: OPM2 (28.3%), MM1.S (11.7%), and H929 (7.77%). The actual MM lysis rate is greater than the proportion of CD24⁺ cells, indicating that CD24⁻ cells are also being killed. We thought this was due to the Bystander Effects of CAR-T cells. The term bystander effect refers to the phenomenon where CAR T-cells not only target

cancer cells expressing a specific antigen but also stimulate cytotoxic activity against nearby cancer cells that lack the targeted antigen expression^{28, 29}. Because our in vitro cytotoxicity assays were conducted in the 96-well plates, when CAR-T cells target CD24⁺ MM, CAR-T cells were activated and released a large amount of cytokines (such as IFN- γ , IL-2, TNF- α , perforin, and granzyme) to kill MM cells. Through our measurements, the concentrations of IFN- γ , IL-2, and TNF- α in the culture medium were significantly increased. CD24⁻ MM in the same environment is also killed by these cytokines released by CAR-T cells, and the higher the cytokine concentration, the greater the MM lysis rate. We apologize for not providing detailed explanation of this phenomenon. We have included this important information in the revised discussion.

Moreover, the reviewer suggests that the use of multiple myeloma cell lines with CD24 knockout (CD24KO) could provide a more rigorous test of the CAR's specificity. Thanks for the reviewer's valuable suggestion. In the revised version, we conducted CRISPR-Cas9-mediated CD24 knockout in OPM2 and MM1.S cells. Subsequently, we performed cytotoxicity assays and CAR-T cell activation experiments using these two knockout cells. From the results shown in the figure below, it is evident that after knockout, the surface expression of CD24 on OPM2 and MM1.S cells significantly decreased, with the proportion of CD24⁺ cells in both cell types reduced to below 2%. When the ratio of CD24-CAR-T cell : MM cell was 5:1, 13.7% of target cells were lysed in the OPM2^{CD24KO} cell line, and there was a slight increase compared to the MOCK-CAR-T group (9.58%), the difference was not statistically significant. Similarly, the same phenomenon occurred when MM1.S^{CD24KO} cell line was used as target cells. Meanwhile, CD24-CAR-T cells also didn't show the highest activation when OPM2^{CD24KO} and MM1.S^{CD24KO} cell lines were used as target cells. Compared to the T cells and MOCK-CAR-T cells, CD24-CAR-T cells exhibited a slight increase in the production of IFN- γ , IL-2, and TNF- α , but the differences were not significant. These results further substantiate the specificity of CD24-CAR-T cells targeting.

Please refer to the following:

In revised Supplementary Fig. 7,

After Revision:

Supplementary Fig. 7. CD24-CAR-T cells can't eliminate CD24-knockout MM cells in vitro.

(a) Flow cytometry analysis of CD24 expression in CD24-knockout (CD24KO) MM cells. **(b)**

CAR-T or T cells were added to CD24KO MM cell lines: OPM2^{CD24KO} and MM1.S^{CD24KO} at the effector/target (E/T) ratio from 1:5 to 5:1. After 24 hours of coculture, cytolytic activity was measured (n = 3 independent experiments). (c-d) CD69 expression, IFN- γ , TNF- α and IL-2 concentrations was detected at the E/T ratio was 5:1 after 24 hours of coculture (n = 3 independent experiments). One-way ANOVA was used for statistical analysis. ns = $P > .05$.

In “RESULTS” section, “CD24-CAR-T cells show efficient MM cells killing in vitro” part, page 8 - 9, lines 189 - 200.

After Revision:

Furthermore, we employed CRISPR-Cas9 technology to knock out CD24 in OPM2 and MM1.S cells (Supplementary Fig. 7a). Subsequently, the cytotoxicity of CD24-CAR-T cells against these knockout MM cells was tested. When the ratio of CD24-CAR-T cell : MM cell was 5:1, 13.7% of target cells were lysed in the OPM2^{CD24KO} cell line, a slight increase compared to the MOCK-CAR-T group (9.58%), that was not statistically significant. Similarly, the same phenomenon occurred when MM1.S^{CD24KO} cell line was used as target cells (Supplementary Fig. 7b). Meanwhile, CD24-CAR-T cells also didn't show the highest activation when OPM2^{CD24KO} and MM1.S^{CD24KO} cell lines were used as target cells (Supplementary Fig. 7c). Compared to the T cells and MOCK-CAR-T cells, CD24-CAR-T cells exhibited a slight increase in the production of IFN- γ , IL-2, and TNF- α , but the differences were not even significant (Supplementary Fig. 7d). These results further substantiate the specificity of CD24-CAR-Tcells targeting.

In “DISCUSSION” section, page 17 - 18, lines 417 - 428.

In our in vitro cytotoxicity experiments, we observed an interesting phenomenon. At the ratio 5 : 1 of CD24-CAR-T versus MM cells, the percentage of lysed MM cells were 78.9%, 65.0% and 38.7% in the OPM2, MM1.S and H929 cell lines, respectively (Fig. 2c). The CD24⁺ proportions for the three cell lines were: OPM2 (28.3%), MM1.S (11.7%), and H929 (7.77%)

(Supplementary Fig. 4). The actual MM lysis rate is greater than the proportion of CD24⁺ cells, suggesting that CD24⁻ cells are also being killed. This could be due to bystander effects of CAR-T cells. The term bystander effect refers to the phenomenon where CAR T-cells not only target cancer cells expressing a specific antigen but also stimulate cytotoxic activity against nearby cancer cells that lack the targeted antigen expression^{28, 29}. Because our in vitro cytotoxicity assays were conducted in 96-well plates, when CAR-T cells targeted CD24⁺ MM, CAR-T cells were activated and released large number of cytokines (such as IFN- γ , IL-2, TNF- α , perforin, and granzyme) to kill MM cells.

In “METHODS” section, “CD24 knockout” part, page 22, lines 538 - 548.

After Revision:

CD24 knockout

The lentiCRISPR v2 plasmid was kindly provided by Feng Zhang (Addgene plasmid # 52961).³⁰ To create the lentiCRISPR v2-CD24 plasmid, we synthesized two gRNA sequences targeting exon 1 and exon 2 of CD24, as follows: gRNA1: 5'-AGGGCCTCACCTGCGTGGGT-3' and gRNA2: 5'-ATTTGGGGCCAACCCAGAGT-3'. These gRNA sequences were integrated into the lentiCRISPR v2 vector using BsmBI restriction sites. Subsequently, the lentiCRISPR v2-CD24 plasmids, along with psPAX2 and pVSV-G plasmids, were co-transfected into HEK293T cells using Lipofectamine2000 (Invitrogen). Lentivirus was collected at transduced MM cells. The transduced cells were selectively cultured using 2.5 μ g/ml puromycin (Sigma, USA). Following one week of puromycin selective cultivation, the selected myeloma cells were individually cloned using limited dilution. The knockout of CD24 was confirmed by flow cytometry.

2. The authors use headless CAR as controls, which may be adequate in certain settings but not in all (see 3)

3. A major problem I see in the experiments investigating macrophage phagocytosis in vitro and in vivo is the lack of a control CAR actually killing MM cells efficiently but not addressing macrophage function. One could easily imagine that dying cells by CAR killing could be more easily phagocytosed. In fact other groups have implied macrophage reprogramming in CAR efficiency. This must be properly demonstrated to showcase the mechanistic advance here.

The reviewer raises a very important point. The concerns are entirely valid, specifically regarding the enhanced susceptibility of cells killed by CAR-T to phagocytosis. The reviewer suggested that a control CAR, which efficiently kills MM cells but does not address macrophage function, should be included to provide a more comprehensive assessment of the mechanisms involved. In the subsequent experiments, we included a control CAR-T: BCMA-CAR-T which has potent anti-MM cell activity but does not target CD24.

In vitro phagocytosis was performed by coculture of OPM2, MM1.S, or H929 cells that expressed GFP, DiD stained macrophages, and different CAR-T cells at a ratio of 2:1:1 for 4 hours in the 37 °C incubator. Then, cells were washed with PBS twice, and GFP⁺DiD⁺ cells were detected by flow cytometry and fluorescence microscope. The results showed that compared with the MOCK-CAR-T cells and BCMA-CAR-T cells, the CD24-CAR-T cells strongly promoted phagocytic clearance by macrophages. Flow cytometry also showed that the CD24-CAR-T cells significantly increased the phagocytic clearance by macrophages.

This is a very interesting result. Since we used a 1:2 ratio of CD24-CAR-T cells to MM cells, based on the previous findings (Fig. 5c), there were still viable MM cells in the co-culture system. The role of CD24-CAR-T cells is to enhance macrophage phagocytosis of these surviving cells. While some cells were killed by CAR-T cells, this fraction of dead cells does not emit GFP fluorescence because fluorescence is lost when the cells die.³¹ The figures below represent MM cells that have just been phagocytosed by macrophages, and they are still detectable by GFP signal. Therefore, in the BCMA-CAR-T group, we did not observe a

significant number of phagocytosed MM cells, even though BCMA-CAR-T had a stronger cytotoxic effect on MM cells compared to CD24-CAR-T.

Please refer to the following:

In revised Supplementary Fig. 16,

After Revision:

Supplementary Fig. 16. CD24-CAR-T and Bi-CAR-T cells promote phagocytic clearance by macrophages in vitro. (a) Phagocytosis was performed by coculture of MM1.S cells that expressed GFP (green), DiD-stained macrophages (red), and CAR-T cells at a ratio of 2:1:1. After a 4-hour coculture, suspended cells were washed and detected. Fluorescent images of phagocytic clearance (n = 3 independent experiments). (b) Bar plot showing the percentage of MM1.S phagocytosis detected by flow cytometry analysis (n = 3 independent experiments). (c) Phagocytosis was performed by coculture of H929 cells (n = 3 independent experiments). (d) Bar plot showing the percentage of H929 phagocytosis detected by flow cytometry analysis (n = 3 independent experiments). One-way ANOVA was used for statistical analysis * $P < .05$, ** $P < .01$, *** $P < .001$, ns = $P > .05$.

In “RESULTS” section, “Bispecific CAR-T cells exhibit increased cytolytic activity in vitro” part, page 14, lines 341 - 344.

After Revision:

Compared with the MOCK-CAR-T cells and BCMA-CAR-T cells, the Bi-CAR-T cells strongly promoted phagocytic clearance by macrophages, similar to the CD24-CAR-T cells (Fig. 5h and Supplementary Fig. 16a, c). Flow cytometry also showed the Bi-CAR-T cells significantly increased the phagocytic clearance by macrophages (Fig. 5i and Supplementary Fig. 16b, d).

In “METHODS” section, “Macrophage phagocytosis” part, page 23, lines 562 - 564.

After Revision:

In vitro phagocytosis was performed by coculture of OPM2, MM1.S, or H929 cells that expressed GFP, DiD stained macrophages, and CAR-T cells at a ratio of 2:1:1 for 4 hours in the 37 °C incubator.

4. Activity of dual CAR over single CAR is not overly convincing in vitro, while very convincing in vivo (which is also more relevant), this must be better contrasted in the text.

The reviewer raises a valuable point. In the in vitro experiments, there was not much difference between dual CAR and single CAR (especially BCMA-CAR-T). However, in the in vivo experiments, dual CAR demonstrated a significantly more effective tumor suppression effect. We thought that as we mentioned in response to the first question, the better in vitro inhibition of MM is due to the Bystander Effects of CAR-T cells. BCMA⁺ or CD24⁺ MM cells were also killed by these cytokines released by CAR-T cells. However, in the multiple myeloma mouse models, the overall environment is significantly different from in vitro conditions. Unlike solid tumors that are located in specific areas, multiple myeloma cells are growing in clusters but are also distributed throughout the bone marrow, potentially reducing the impact of Bystander Effects. On the other hand, the escapes of BCMA⁻ or CD24⁻ cells after the single CAR-T treatment were amplified over time in vivo. We observed that in the early stages of single CAR-T treatment in MM1.S and OPM2-xenograft mouse model models, there was a good response (Day 7 - Day 21). However, myeloma relapsed in the later stages of single CAR-T treatment. Dual CAR, which targets both BCMA and CD24 simultaneously, provided better inhibition of tumor relapse.

We apologize for not providing detailed explanation of this phenomenon. We have included this important information in the revised discussion.

Please refer to the following:

In "DISCUSSION" section, page 18, lines 429 - 442.

Another interesting phenomenon was that in our in vitro experiments, there was not much difference between Bi-CAR-T and single-CAR-T (especially BCMA-CAR-T) (Fig. 5c). However, in the in vivo experiments, Bi-CAR-T demonstrated a significantly more effective tumor killing effect (Fig. 6). The better in vitro inhibition of MM cells could be due to bystander effects. The

BCMA- or CD24- MM cells were also killed by these cytokines released by CAR-T cells. However, in the multiple myeloma mouse models, the overall environment is significantly different from in vitro conditions. Unlike solid tumors that are located in specific areas, multiple myeloma cells grow in large clusters, but are also distributed throughout the bone marrow, which may potentially reduce the impact of bystander effects. On the other hand, the escape of BCMA- or CD24- MM cells after the single CAR-T treatment were amplified over time in vivo. In the early stages of single CAR-T treatment in MM1.S and OPM2-xenograft mouse model models, there was a good response (day 7 to day 21). However, myeloma relapsed in the later stages of single CAR-T treatment. Dual CAR, which targets both BCMA and CD24 simultaneously, should provide better inhibition of tumor relapse.

Formal aspects

1. How was cell line identity ascertained?

The reviewer raises an important point. Verifying the authenticity of the cells is essential to uphold the reproducibility and validity of the study's findings. In this study, all cell lines' authenticity was confirmed through STR (Short Tandem Repeat) profiling, and routine mycoplasma testing was performed using a mycoplasma detection kit. We have added this information in the revised version.

Please refer to the following:

In "METHODS" section, "Patient samples and cell lines" part, page 20, lines 495 - 497.

All cell lines' authenticity was confirmed through STR (Short Tandem Repeat) profiling, and routine mycoplasma testing was performed using a mycoplasma detection kit.

2. It is difficult to follow in the figures the amount of replicates and independent repeats performed and where data was pooled. This must be made crystal clear for each and every figure and subfigure.

Thanks for the reviewer's valuable suggestion. This information is essential for readers to understand the robustness and reliability of the findings. We apologize for not providing this information. We have included this important information in the revised Figure legends and Supplemental Figure legends.

We wish to thank the reviewers for their comments, which led to a number of important changes in the manuscript. We believe the new version is substantially improved because of that.

With many thanks, on behalf of all authors,

REFERENCES

1. Sung MH, *et al.* Dynamic effect of bortezomib on nuclear factor-kappaB activity and gene expression in tumor cells. *Mol Pharmacol* **74**, 1215-1222 (2008).
2. Van Waes C. Nuclear factor-kappaB in development, prevention, and therapy of cancer. *Clin Cancer Res* **13**, 1076-1082 (2007).
3. Hideshima T, *et al.* Bortezomib induces canonical nuclear factor-kappaB activation in multiple myeloma cells. *Blood* **114**, 1046-1052 (2009).
4. Rosenberg AS. From mechanism to resistance - changes in the use of dexamethasone in the treatment of multiple myeloma. *Leuk Lymphoma* **64**, 283-291 (2023).
5. Nissen RM, Yamamoto KR. The glucocorticoid receptor inhibits NFkappaB by interfering with serine-2 phosphorylation of the RNA polymerase II carboxy-terminal domain. *Genes Dev* **14**, 2314-2329 (2000).
6. Ju JH, *et al.* CD24 enhances DNA damage-induced apoptosis by modulating NF-kappaB signaling in CD44-expressing breast cancer cells. *Carcinogenesis* **32**, 1474-1483 (2011).
7. Wang X, *et al.* CD24-Siglec axis is an innate immune checkpoint against metaflammation and metabolic disorder. *Cell Metab* **34**, 1088-1103 e1086 (2022).
8. Fink EC, Ebert BL. The novel mechanism of lenalidomide activity. *Blood* **126**, 2366-2369 (2015).
9. Kotla V, *et al.* Mechanism of action of lenalidomide in hematological malignancies. *J Hematol Oncol* **2**, 36 (2009).
10. Gao M, *et al.* Identification and Characterization of Tumor-Initiating Cells in Multiple Myeloma. *J Natl Cancer Inst* **112**, 507-515 (2020).
11. Klapdor R, *et al.* Characterization of a Novel Third-Generation Anti-CD24-CAR against Ovarian Cancer. *Int J Mol Sci* **20**, (2019).

12. Sterner RC, Sterner RM. CAR-T cell therapy: current limitations and potential strategies. *Blood Cancer J* **11**, 69 (2021).
13. Brudno JN, Kochenderfer JN. Recent advances in CAR T-cell toxicity: Mechanisms, manifestations and management. *Blood Rev* **34**, 45-55 (2019).
14. Jackson D, Waibel R, Weber E, Bell J, Stahel RA. CD24, a signal-transducing molecule expressed on human B cells, is a major surface antigen on small cell lung carcinomas. *Cancer Res* **52**, 5264-5270 (1992).
15. Elghetany MT, Patel J. Assessment of CD24 expression on bone marrow neutrophilic granulocytes: CD24 is a marker for the myelocytic stage of development. *Am J Hematol* **71**, 348-349 (2002).
16. Calaora V, Chazal G, Nielsen PJ, Rougon G, Moreau H. mCD24 expression in the developing mouse brain and in zones of secondary neurogenesis in the adult. *Neuroscience* **73**, 581-594 (1996).
17. Shirasawa T, Akashi T, Sakamoto K, Takahashi H, Maruyama N, Hirokawa K. Gene expression of CD24 core peptide molecule in developing brain and developing non-neural tissues. *Dev Dyn* **198**, 1-13 (1993).
18. Cram DS, McIntosh A, Oxbrow L, Johnston AM, DeAizpurua HJ. Differential mRNA display analysis of two related but functionally distinct rat insulinoma (RIN) cell lines: identification of CD24 and its expression in the developing pancreas. *Differentiation* **64**, 237-246 (1999).
19. Fang X, Zheng P, Tang J, Liu Y. CD24: from A to Z. *Cell Mol Immunol* **7**, 100-103 (2010).
20. Velasco R, Mussetti A, Villagran-Garcia M, Sureda A. CAR T-cell-associated neurotoxicity in central nervous system hematologic disease: Is it still a concern? *Front Neurol* **14**, 1144414 (2023).
21. Philip B, *et al.* A highly compact epitope-based marker/suicide gene for easier and safer T-cell therapy. *Blood* **124**, 1277-1287 (2014).
22. Roybal KT, *et al.* Engineering T Cells with Customized Therapeutic Response Programs Using Synthetic Notch Receptors. *Cell* **167**, 419-432 e416 (2016).
23. Li W, *et al.* Identification of potential resistance mechanisms and therapeutic targets for the relapse of BCMA CAR-T therapy in relapsed/refractory multiple myeloma through single-cell sequencing. *Exp Hematol Oncol* **12**, 44 (2023).

24. Krautter F, *et al.* Characterisation of endogenous Galectin-1 and -9 expression in monocyte and macrophage subsets under resting and inflammatory conditions. *Biomed Pharmacother* **130**, 110595 (2020).
25. Hu C, *et al.* CellMarker 2.0: an updated database of manually curated cell markers in human/mouse and web tools based on scRNA-seq data. *Nucleic Acids Res* **51**, D870-D876 (2023).
26. Zavidij O, *et al.* Single-cell RNA sequencing reveals compromised immune microenvironment in precursor stages of multiple myeloma. *Nat Cancer* **1**, 493-506 (2020).
27. Gai D, *et al.* CST6 suppresses osteolytic bone disease in multiple myeloma by blocking osteoclast differentiation. *J Clin Invest* **132**, (2022).
28. Upadhyay R, *et al.* A Critical Role for Fas-Mediated Off-Target Tumor Killing in T-cell Immunotherapy. *Cancer Discov* **11**, 599-613 (2021).
29. Klampatsa A, Leibowitz MS, Sun J, Liousia M, Arguiri E, Albelda SM. Analysis and Augmentation of the Immunologic Bystander Effects of CAR T Cell Therapy in a Syngeneic Mouse Cancer Model. *Mol Ther Oncolytics* **18**, 360-371 (2020).
30. Sanjana NE, Shalem O, Zhang F. Improved vectors and genome-wide libraries for CRISPR screening. *Nat Methods* **11**, 783-784 (2014).
31. Lowder M, Unge A, Maraha N, Jansson JK, Swiggett J, Oliver JD. Effect of starvation and the viable-but-nonculturable state on green fluorescent protein (GFP) fluorescence in GFP-tagged *Pseudomonas fluorescens* A506. *Appl Environ Microbiol* **66**, 3160-3165 (2000).

REVIEWER COMMENTS

Reviewer #1 (Remarks to the Author):

The authors addressed previous comments and manuscript can be accepted in the present form.

Reviewer #2 (Remarks to the Author):

The authors adequately answered our comments in the revised version. There are no further comments from our side.

Reviewer #3 (Remarks to the Author):

Reviewer #4 (Remarks to the Author):

I greatly appreciate the authors efforts in answering my comments for reviewing manuscript NCOMMS-23-31192, which improved the bioinformatics analysis part of this comprehensive work. Please find below my comments based on manuscript version NCOMMS-23-31192A:

Q1: The authors reasoned that their scRNA-seq processing pipeline successfully eliminated potential doublet cells and further stated that these cells are true cells and provided independent evidence from public databases, showing some B/plasma cells expressed signals of CD3D expression. While I appreciate the additional work done by authors, I am not fully convinced by the current response. I have the following additional comments to make:

1. The authors need to clearly specify what's the methods/tools/strategies they used to remove doublets. The current description is indirect.
2. Could the authors provide the absolute number and the fraction of CD3D positive B/plasma cells?
3. In addition to CD3D, could the authors check if they also express CD3E and/or CD3G?
4. Any chances those are proliferative cells with mixed lineage features? Could the authors check the expression of canonical proliferation markers like MKI67/CENPF/TYMS/TOP2A etc according to the Seurat tutorial (https://satijalab.org/seurat/articles/cell_cycle_vignette.html)?
5. Did the author checked the cell metrics related to doublets? Like number of genes/transcripts detected per cell? Did the dual positive cells show obvious different ngene/nfeature compared with other cells?
6. Could the authors provide further validation using perpendicular platforms like FACs for mIF to show the existence of CD3D positive B/plasma cells at protein level? If the authors speculation is true, this will be a very interesting point to pursue further (in future work) considering that there have been previous reports showing dual positive cells were associated with specific conditions like for example CAR transduction (ref: PMID32929049) or disease conditions (ref: PMID34341278, ref: PMID19011566).

Q2: I appreciate the additional analysis done by the reviewers which clearly answered my questions and thus have no further question.

Q3: I have no further question.

Q4: I have no further question.

Q5: No further question and I appreciate the additional work done by the authors to confirm this conclusion.

Q6: no further question

Q7: I appreciate the additional work done by authors to answer this question. Additionally, I wonder if the authors observed potential enhancement of cell communications (that are involved in "don't eat me" signal) in treated group compared with ctrl group?

Q8: I have no further question.

Q9: Please confirm if the plot shown in the answer is PCA plot or the UMAP dimension reduction plot. They are technically different. People typically use metrics like silhouette score, to quantify the effect of batches. Could the authors provide the quantification results that batch effect was not involved? One potential way is to evaluate the silhouette score differences before and after batch correction (if the scRNA-seq data was generated in different batches), as is shown in Supp figure 1a from PMID:37248301.

Minor Q1: I greatly appreciate the additional work done by the authors to answer this minor comment. I have no additional question.

Reviewer #5 (Remarks to the Author):

For the avoidance of doubts, I am exclusively commenting on the points of previous reviewer #5. My points have been very well addressed by the authors.

We sincerely appreciate the five reviewers recognizing our revised manuscript. We also extend our gratitude to Reviewer #4 for providing additional helpful suggestions. Appended below are the Reviewer #4's comments. Our point-by-point responses have been inserted, including descriptions of changes made in the amended manuscript. The reviewers' comments are indicated in blue, while our responses are indicated in black. The revised parts in the manuscript are highlighted in yellow.

Reviewer #4

I greatly appreciate the authors efforts in answering my comments for reviewing manuscript NCOMMS-23-31192, which improved the bioinformatics analysis part of this comprehensive work. Please find below my comments based on manuscript version NCOMMS-23-31192A:

We appreciate the reviewer's positive evaluation of our revision efforts. Through the questions raised by the reviewer, we have conducted a more in-depth exploration of the quality control issues related to single-cell sequencing data. The methods section has been revised to provide a detailed description of the quality control procedures. Additionally, we have validated the concerns raised by reviewer #4 regarding CD3-positive plasma/MM cells using flow cytometry. In the following responses, we will address the reviewers' questions carefully to enhance the rigor of our manuscript.

Q1: The authors reasoned that their scRNA-seq processing pipeline successfully eliminated potential doublet cells and further stated that these cells are true cells and provided independent evidence from public databases, showing some B/plasma cells expressed signals of CD3D expression. While I appreciate the additional work done by authors, I am not fully convinced by the current response. I have the following additional comments to make:

1. The authors need to clearly specify what's the methods/tools/strategies they used to remove doublets. The current description is indirect.

Thanks for the valuable suggestion from the reviewer. We apologize for not including specific information about Quality Control in the methods section. Following the reviewer's suggestion, we have provided a more detailed description of the quality control and data filtering processes for scRNA analysis.

Please refer to the following:

In "METHODS" section, "Patients Single-cell RNA sequencing (scRNA-seq) data process" part, page 20, lines 476 - 498.

The raw scRNA-seq data were loaded into R through the Seurat V4 package. The Seurat's integration workflows were used for quality control. First, cells exhibiting low-complexity libraries, indicating detection of transcripts aligned to fewer than 200 genes, potentially representing dying or apoptotic cells (more than 10% of unique molecular identifiers stemming from mitochondrial genes), as well as cells with high-complexity libraries (with detected transcripts aligned to more than 7,000 genes), were excluded. **We used DoubletFinder function to identify potential doublets within our dataset. Essentially, a doublet is characterized as a single-cell library that represents more than one cell. Upon closer inspection of certain known markers, it was observed that the implicated cluster comprises doublets of more than one cell type, as no cell type is recognized for robustly expressing both markers simultaneously. Doublets were individually removed from each sample, employing an anticipated doublet rate of 0.05.** Subsequently, the cell expression matrix of each sample underwent normalization using the NormalizeData function with default parameters. Following this, the FindVariableFeatures function, with default parameters, was

employed to identify highly variable genes (HVGs) within each normalized matrix. The `SelectIntegrationFeatures` function was then utilized, specifying `nfeatures = 2,000`, to select genes for the integration of multiple samples. To mitigate the impact of the cell cycle on data integration, we excluded cell-cycle-related genes from the gene set. Sequentially, the `ScaleData` and `RunPCA` functions were applied with the parameter `features` set to these selected genes. Subsequent to scaling the matrix for each dataset, PCA was conducted. We employed the `FindIntegrationAnchors` function, specifying `reduction = 'rpca'`, to identify a set of anchors between all matrices. These anchors were used to integrate the matrices through the `IntegrateData` function with parameter `dims = 1:50`. Finally, the `ScaleData` function was applied to scale the integrated matrix using default parameters.

2. Could the authors provide the absolute number and the fraction of CD3D positive B/plasma cells?

The reviewer raised a valid point. The cell proportion provides the most straightforward information for these CD3D-positive B/plasma cells. Through calculation, we determined that the fraction of CD3D-positive B/plasma cells to the total B/plasma cells is 1.7% (255/14,771). This proportion closely aligns with the 0.2%-2.8% ratio observed in the flow cytometry analysis of six myeloma patients. For more details, please refer to the response to Question 6.

3. In addition to CD3D, could the authors check if they also express CD3E and/or CD3G?

The reviewer raised a valuable point. Through comprehensive analysis of CD3E and CD3G, we can ascertain whether these cells truly express CD3. From the UMAP below, it is evident that CD3E and CD3G are also expressed in this cell population. By calculation, we determined that CD3E-positive and CD3G-positive B/plasma cells account for 2.7% (406/14,771) and 0.9% (138/14,771) of the total B/plasma cells, respectively.

4. Any chances those are proliferative cells with mixed lineage features? Could the authors check the expression of canonical proliferation markers like MKI67/CENPF/TYMS/TOP2A etc according to the Seurat tutorial (https://satijalab.org/seurat/articles/cell_cycle_vignette.html)?

Thanks for the valuable suggestion from the reviewer. And thank reviewer for providing the Seurat tutorial. Through the analysis, We found that the expression level of proliferation markers MKI67/CENPF/TYMS/TOP2A in CD3D-positive plasma/MM cells is relatively low. Please see the figure below. We infer that CD3D-positive plasma/MM cells are not proliferative cells.

5. Did the author checked the cell metrics related to doublets? Like number of genes/transcripts detected per cell? Did the dual positive cells show obvious different ngene/nfeature compared with other cells?

Thanks for the valuable suggestion from the reviewer. In this cluster, we compared the ngene/nfeature between CD3D-positive and CD3D-negative cells. We found that there was no significant difference in the ngene/nfeature between CD3D-positive and CD3D-negative cells. Please see the figure below.

6. Could the authors provide further validation using perpendicular platforms like FACs for mIF to show the existence of CD3D positive B/plasma cells at protein level? If the authors speculation is true, this will be a very interesting point to pursue further (in future work) considering that there have been previous reports showing dual positive cells were associated with specific conditions like for example CAR transduction (ref: PMID32929049) or disease conditions (ref: PMID34341278, ref: PMID19011566).

Thanks for the valuable suggestion from the reviewer. We conducted a flow cytometry analysis of CD3 expression in plasma/MM cells from six myeloma patients. We found that the plasma/MM cells in these six myeloma patients expressed CD3, with expression levels ranging from 0.2% to 2.8%. Please see the figure below. We validated this at the cell surface protein level, confirming the presence of a subset of plasma/MM cells expressing CD3. This observation aligns with some literature. ^{1, 2, 3}

As of current literature, CD3-positive B/plasma cells have not received widespread attention.

We are grateful to the reviewers for providing a novel research direction for the next steps of our study, and we plan to incorporate this suggestion into our future investigations.

Figure legend: Flow cytometry analysis of CD3 expression in CD138⁺ plasma/MM cells.

Q7: I appreciate the additional work done by authors to answer this question. Additionally, I wonder if the authors observed potential enhancement of cell communications (that are involved in “don’t eat me” signal) in treated group compared with ctrl group?

The reviewer raises a valuable point. We found in the PBS and MOCK-CAR-T treatment group, that the Siglec-G (known as Siglec-10 in Humans) signaling pathway showed a significant enhancement. In contrast, in the CD24-CAR-T treatment group, this signal did not significantly enhance. The interaction between CD24 expressed on tumors and Siglec-G is crucial for

activating the "don't eat me" signal pathway. CD24-CAR-T cells can effectively inhibit the binding of CD24 to Siglec-G, thereby inhibiting the "don't eat me" signal.

Please refer to the following:

In revised Supplementary Fig. 13d,

After Revision:

Supplementary Fig. 13. Cell-cell communication analysis. (d) Hierarchy plots showing the Siglec-G communication signal in PBS and MOCK-CAR-T treatment groups.

In "RESULTS" section, "CD24-CAR-T cells modulate myeloma microenvironment" part, page 13, lines 308 - 318.

After Revision:

In the CD24-CAR-T treatment group, the most pronounced communication signals were related to the Galectin-9 (Lgals9) signaling pathway. The most critical sources and targets of these signals were neutrophils and monocytes (Supplementary Fig. 13c). It has been reported that Galectin-9 is upregulated in murine M1 macrophages, ⁴ which aligns with our flow cytometry results indicating a significant increase in M1 macrophages after CD24-CAR-T treatment. We

found in the PBS and MOCK-CAR-T treatment groups, that the Siglec-G (known as Siglec-10 in Humans) signaling pathway showed a significant enhancement (Supplementary Fig. 13d). In contrast, in the CD24-CAR-T treatment group, this signal did not significantly enhance. The interaction between CD24 and Siglec-G is crucial for activating the "don't eat me" signal pathway.⁵ CD24-CAR-T cells can effectively inhibit the binding of CD24 to Siglec-G, thereby inhibiting the "don't eat me" signal.

Q9: Please confirm if the plot shown in the answer is PCA plot or the UMAP dimension reduction plot. They are technically different. People typically use metrics like silhouette score, to quantify the effect of batches. Could the authors provide the quantification results that batch effect was not involved? One potential way is to evaluate the silhouette score differences before and after batch correction (if the scRNA-seq data was generated in different batches), as is shown in Supp figure 1a from PMID:37248301.

Thanks for the valuable suggestion from the reviewer. The reviewer is correct. The plot we presented in our previous response is a UMAP. We apologize for the technical confusion in our previous response. In this study, we used a Seurat rPCA approach as a batch correction method. Following the reviewer's suggestion, we used the silhouette coefficient score to quantify the effect of batch correction for the Neutrophils cluster.⁶ We chose this cluster for presentation because it is the major cell type in our data (similar to silhouette scores shown for major cell types in Supp Figure 1a from PMID: 37248301).⁷ Silhouette coefficients score measures the similarity of a cell to its own cluster compared to other clusters. This yields a score ranging from -1 to +1, with a higher score indicating better performance.⁶ The silhouette coefficients score considers aspects of both sample mixing and local structure. We observed a significant increase in the silhouette coefficients score after batch correction. Please see the figure below. This demonstrates the effectiveness of our batch correction.

Figure legend: Box plots showing the silhouette coefficients scores for major cell type Neutrophils without (raw) and with batch effect correction.

We wish to thank the reviewer for the comments, which led to a number of important changes in the manuscript. We believe the new version is improved because of that.

With many thanks, on behalf of all authors,

REFERENCES

1. Yagci M, Sucak GT, Akyol G, Haznedar R. Hepatic failure due to CD3+ plasma cell infiltration of the liver in multiple myeloma. *Acta Haematol* **107**, 38-42 (2002).
2. Spier CM, *et al.* T-cell antigen-positive multiple myeloma. *Mod Pathol* **3**, 302-307 (1990).
3. Tang YL, Chau CY, Yap WM, Chuah KL. CD3 expression in plasma cell neoplasm (multiple myeloma): a diagnostic pitfall. *Pathology* **44**, 668-670 (2012).
4. Krautter F, *et al.* Characterisation of endogenous Galectin-1 and -9 expression in monocyte and macrophage subsets under resting and inflammatory conditions. *Biomed Pharmacother* **130**, 110595 (2020).
5. Barkal AA, *et al.* CD24 signalling through macrophage Siglec-10 is a target for cancer immunotherapy. *Nature* **572**, 392-396 (2019).
6. Stuart T, *et al.* Comprehensive Integration of Single-Cell Data. *Cell* **177**, 1888-1902 e1821 (2019).
7. Chu Y, *et al.* Pan-cancer T cell atlas links a cellular stress response state to immunotherapy resistance. *Nat Med* **29**, 1550-1562 (2023).

REVIEWERS' COMMENTS

Reviewer #4 (Remarks to the Author):

Q1: no further comments.

Q2: thanks for the authors to providing additional quantification metrics for these dual positive cells. I have no further comments.

Q3: no further comments.

Q4: no further comments.

Q5: no further comments.

Q6: I appreciate the additional validation data. One minor suggestion is to discuss these data in the manuscripts' discussion part.

Q7: no further comments.

Q9: I have concerns for the authors response to this question: Please clarify If batch correction methods was applied in the scRNA-seq analysis.

We extend our gratitude to Reviewer #4 for providing additional helpful minor suggestions. Appended below are the Reviewer #4's comments. Our point-by-point responses have been inserted, including descriptions of changes made in the amended manuscript. The reviewers' comments are indicated in blue, while our responses are indicated in black. The revised parts in the manuscript are highlighted in yellow.

Reviewer #4

Q6: I appreciate the additional validation data. One minor suggestion is to discuss these data in the manuscripts' discussion part.

This is a valid suggestion. We discussed the potential significance of this small cell population in the Discussion Section.

Please refer to the following:

In "DISCUSSION" section, page 19, lines 448 - 454.

During the analysis of single-cell sequencing data, we identified an intriguing cell population. As shown in the first panel of Supplementary Fig. 1, a small subset of plasma/MM cells expresses CD3D, which we and others have observed similar expression patterns in MM plasma cells and Waldenström's macroglobulinemia (WM) B cells.^{1,2} Currently, it is not clear why these B and plasma cells express T cell marker. One potential possibility is tumor cell dedifferentiation. Given that CAR-T therapies are predominantly manufactured using CD3⁺ T cells, the fate of this cell subset in the CAR production process warrants further investigation in subsequent studies.

Q9: I have concerns for the authors response to this question: Please clarify If batch correction methods was applied in the scRNA-seq analysis.

The reviewer raised a valid point. Yes, we now made a clear description about batch correction methods applied in the scRNA-seq analysis.

Please refer to the following:

In “METHODS” section, “Patients Single-cell RNA sequencing (scRNA-seq) data process” part, page 21, lines 502 - 509.

Sequentially, the RunPCA and ScaleData functions were applied with the parameter features set to these selected genes. Subsequent to scaling the matrix for each dataset, PCA was conducted. To reduce batch effects, we applied the “anchor” integration method (functions FindIntegrationAnchors and IntegrateData).³ We employed the FindIntegrationAnchors function, specifying reduction = ‘rpca,’ to identify a set of anchors between all matrices. These anchors were used to integrate the matrices through the IntegrateData function with parameter dims = 1:50. Finally, the ScaleData function was applied to scale the integrated matrix using default parameters.

We wish to thank the reviewer for the comments, which led to a number of important changes in the manuscript. We believe the new version is improved because of that.

With many thanks, on behalf of all authors,

REFERENCES

1. Yagci M, Sucak GT, Akyol G, Haznedar R. Hepatic failure due to CD3+ plasma cell infiltration of the liver in multiple myeloma. *Acta Haematol* **107**, 38-42 (2002).
2. Hao M, *et al.* Gene Expression Profiling Reveals Aberrant T-cell Marker Expression on Tumor Cells of Waldenstrom's Macroglobulinemia. *Clin Cancer Res* **25**, 201-209 (2019).
3. Stuart T, *et al.* Comprehensive Integration of Single-Cell Data. *Cell* **177**, 1888-1902 e1821 (2019).